# L-PINN: A Langevin Dynamics Approach with Balanced Sampling to Improve Learning Stability in Physics-Informed Neural Networks

## Abstract

Physics-informed neural networks (PINNs) have emerged as a promising technique solving partial differential equations (PDEs). However, PINNs face challenges in resource efficiency (e.g., repeatedly sampling of collocation points) and achieving fast convergence to accurate solutions. To address these issues, adaptive sampling methods that focus on collocation points with high residual values have been proposed, enhancing both resource efficiency and solution accuracy. While these high residual-based sampling methods have demonstrated exceptional performance in solving certain stiff PDEs, their potential drawbacks, particularly the relative neglect of points with medium and low residuals, remain under-explored. In this paper, we investigate the limitations of high residual-based methods concerning learning stability as model complexity increases. We provide a theoretical analysis demonstrating that high residual-based methods require tighter upper bound on the learning rate to maintain stability. To overcome this limitation, we present a novel Langevin dynamics-based PINN (L-PINN) framework for adaptive sampling of collocation points, which is designed to improve learning stability and convergence speed. To validate the effectiveness, we evaluated the L-PINN framework against existing adaptive sampling approaches for PINNs. Our results indicate that the L-PINN framework achieves superior relative $L^2$ error performance in solutions while demonstrating faster or comparable convergence stability. Furthermore, we demonstrated that our framework exhibits robust performance across a range of model complexities, indicating its potential for compatibility with larger neural network size in addressing challenging PDEs.

## 1 Introduction

Partial differential equations (PDEs) are crucial for describing various physical phenomenon such as heat transfer (Haghighat et al., 2021, Cai et al., 2021b), flow dynamics (Shi et al., 2021, Jagtap et al., 2022, Nazari et al., 2022), propagation dynamics (Pettit & Wilson, 2020, bin Waheed et al., 2021), optics and epidemiology (Lin & Chen, 2022, Rodríguez et al., 2023). Getting accurate and efficient solutions to PDEs is essential across numerous industries reliant on these descriptions. With advancements in deep learning, physics-informed neural networks (PINNs) have emerged as a promising method for solving PDEs. The training process of collocation-based PINNs involves minimizing total errors, including initial condition (IC), boundary condition (BC), and PDE errors measured at collocation points (Nabian et al., 2021, Zeng et al., 2022, Gao & Wang, 2023, Toloubidokhti et al., 2024, Lau et al., 2024). In particular, IC, BC, and PDE errors are incorporated as soft constraints on experimental data, ensuring that the predicted solutions satisfy these essential requirements.

This collocation-based learning method enhances the capability of PINNs by reducing the need for extensive experimental data collection across spatio-temporal ranges, demonstrating success in various industries as a promising alternative to traditional numerical methods like the finite difference method and finite element method (Zhu et al., 2019, Bar-Sinai et al., 2019, Li et al., 2020). However, collocation-based PINN (hereafter referred to as PINNs) encounter challenges in efficiently setting collocation points within the constraints of a limited sampling budget, and in achieving fast convergence to accurate solutions. A key challenge arises from the presence of small regions with abrupt changes, in contrast to the larger, smoother regions. This issue is particularly evident in stiff

PDEs, which are often characterized by discontinuities, such as sudden transitions or jumps across the spatio-temporal domain.

To address these challenges, two main adaptive sampling approaches have been proposed: residual distribution-based sampling and high residual-based sampling. Residual distribution-based sampling resamples collocation points according to the *residual distribution* at each iteration, ensuring a proportional balance of points based on their residuals. In contrast, high residual-based approaches directly target collocation points with high residuals, replacing those with lower values and often neglecting low-residual regions. This focus can make it difficult to discern the analytical form of the residual distribution. Nonetheless, high residual-based methods have recently demonstrated superior performance. This raises an important question: *Should we then focus exclusively on high residual points in an extreme manner?* Addressing this unresolved issue requires a thorough analysis of the trade-offs and risks involved in different adaptive sampling strategies.

In this paper, we respond to this question by presenting a theoretical analysis that highlights the importance of balanced adaptive sampling. We propose a novel Langevin dynamics-based PINN (L-PINN) framework for balanced adaptive sampling of collocation points, ensuring an continuous sampling process. We evaluated our L-PINN framework against existing adaptive sampling approaches for PINNs, demonstrating consistently reliable relative $L^2$ error rates and robust convergence stability. Furthermore, our framework adapts well to varying learning rates, highlighting its robustness across different training configurations. Notably, our proposed framework performs effectively across in diverse PDEs, distinguishing itself with enhanced learning stability.

## 2 BACKGROUND AND RELATED WORK

**Physics-informed neural networks.** The basic PINN framework (Raissi et al., 2017) utilizes deep neural networks as function approximators $f_\theta$ to estimate the solution $u$ of a non-linear PDE. The PDE formulation can be defined as follows:

$$u_t + \mathcal{N}_x[u] = 0, \quad x \in \mathcal{X} \subset \mathbb{R}^d, \quad t \in [0, T]; \tag{2.1}$$

$$u(x, 0) = h(x), \quad x \in \mathcal{X} \subset \mathbb{R}^d; \tag{2.2}$$

$$u(x, t) = g(x, t), \quad x \in \partial\mathcal{X} \subset \mathbb{R}^d, \quad t \in [0, T] \tag{2.3}$$

where $u(x, t)$ denotes the hidden solution at spatial and temporal coordinates $x, t$, $\mathcal{N}_x[\cdot]$ is the non-linear differential operator, $\mathcal{X}$ is the spatial domain, $\partial\mathcal{X}$ is the boundary, and $T$ is the time range. The spatio-temporal domain is $\Omega = \mathcal{X} \times [0, T]$, with collocation points $\mathbf{x} = (x, t) \in \Omega$ and spatial dimension $d$. The PDE residuals $\mathcal{R}_\theta(\mathbf{x})$ and loss function on collocation points $\{\mathbf{x}_n^{\text{pde}}\} \subset \Omega$ are calculated as:

$$\mathcal{R}_\theta(\mathbf{x}) = \frac{\partial}{\partial t} f_\theta(\mathbf{x}) + \mathcal{N}_x[f_\theta](\mathbf{x}), \mathbf{x} \in \Omega \tag{2.4}$$

$$\mathcal{L}_{\text{pde}}(\{\mathbf{x}_n^{\text{pde}}\}; \theta) = \mathbb{E}_{\mathbf{x} \sim \mathcal{U}(\Omega)} |\mathcal{R}_\theta(\mathbf{x})|^k \approx \frac{1}{N_{\text{pde}}} \sum_{n=1}^{N_{\text{pde}}} |\mathcal{R}_\theta(\mathbf{x}_n^{\text{pde}})|^k \tag{2.5}$$

where $\mathcal{U}(\Omega)$ is the uniform distribution over $\Omega$ and $N_{\text{pde}}$ represents the number of sample points of PDE loss. Then, in a similar manner, the total loss function $\mathcal{L}$ is defined as:

$$\mathcal{L}(\{\mathbf{x}_n\}; \theta) = \lambda_{\text{pde}}\mathcal{L}_{\text{pde}}(\{\mathbf{x}_n^{\text{pde}}\}; \theta) + \lambda_{\text{ic}}\mathcal{L}_{\text{ic}}(\{\mathbf{x}_n^{\text{ic}}\}; \theta) + \lambda_{\text{bc}}\mathcal{L}_{\text{bc}}(\{\mathbf{x}_n^{\text{bc}}\}; \theta) \tag{2.6}$$

Hyperparameters $\lambda_{\text{pde}}$, $\lambda_{\text{ic}}$, and $\lambda_{\text{bc}}$ control the balance between the PDE, IC, and BC loss terms. Then, $f_\theta$ is trained to estimate appropriate solution $u$ for PDEs by minimizing the total loss $\mathcal{L}$.

**Adaptive sampling based on residual distribution.** Classical PINNs commonly used a uniform distribution sampling strategy for collocation points. To improve this, an adaptive sampling method based on the PDE residuals was proposed (Nabian et al., 2021). In this method, the residual-based distribution for adaptive sampling is calculated by dividing the PDE loss of each collocation point by the arithmetic average of the total PDE loss, i.e., the $n$-th collocation point is sampled with probability $p(\mathbf{x}_n) = \frac{|\mathcal{R}_\theta(\mathbf{x}_n)|^k}{\sum_m |\mathcal{R}_\theta(\mathbf{x}_m)|^k}$ where $\mathcal{R}_\theta(\mathbf{x}_n)$ is the residual at $\mathbf{x}_n \in \Omega$. A more generalized

method, the residual-based adaptive distribution (RAD), was later introduced (Wu et al., 2023). RAD incorporates one additional non-negative hyperparameter $c$ represented by $p(\mathbf{x}) \propto \frac{|\mathcal{R}_\theta(\mathbf{x})|^k}{\mathbb{E}|\mathcal{R}_\theta(\mathbf{x})|^k} + c$. Specifically, $c$ regulate the degree of uniformity in sampling, allowing for a balance between low and high residuals. When $k$ dominates $c$, the sampling probability increases proportionally to the magnitude of the residual favoring high residuals. Conversely, when $c$ dominates $k$, the influence of the residual diminishes, leading to uniform sampling. RAD can adjust the importance of low and high residual points using hyperparameters, depending on the form of the given PDE.

**Adaptive sampling focused on high residuals.** Alongside the residual distribution-based approach, another prominent line of research focuses on sampling methods without directly approximating the underlying distribution. The methods introduced below could be interpreted as special cases of RAD in extreme $k, c$ settings; however, we aim to characterize them based on whether they involve estimating the sampling distribution. One such approach is the high residual-based adaptive refinement (RAR) scheme Lu et al., 2021a, where the top-$M$ high residual collocation points are added to the training batches for PINN models. This process continues until the mean PDE residual satisfies a predefined error tolerance. Although RAR showed remarkable improvement, the number of collocation points could continuously grow without any replacement, leading increased computational complexity. To address this, a retain-resample-release (R3) sampling method (Daw et al., 2023) was proposed to enhance sample efficiency by retaining high residual points, uniformly resampling some points to improve diversity, and releasing collocation points with low residuals. While these method focus on improving sampling strategies, a different question has arisen regarding whether the $L^2$ physics-informed loss is appropriate for Hamilton-Jacobi-Bellman (HJB) equations (Wang et al., 2022a). In response, an adversarial training method has been proposed as an alternative, aiming to optimize the $L^\infty$ norm for solving high-dimensional PDEs. Later, we will describe this in more detail, during the process of conducting adversarial training to leverage $L^\infty$, partial gradient information is utilized. As a result, it can be observed that this leads to a sampling technique that tends to focus on high residual samples.

**Unresolved questions in adaptive sampling methods.** While adaptive sampling methods show promising results, several theoretical aspects remain unclear. In particular, there is a lack of theoretical analysis on the balancing effect. Although many studies report success with adaptive sampling, limited analysis exists on the impact of focusing on high residuals. Specifically, it remains unexplained why algorithms that excessively emphasize high residuals exhibit instability during training.

To address these issues, we first investigate the relationship between learning stability and the degree of emphasis on high residuals with respect to model complexity, and examine how an exclusive focus on high residuals may lead to performance degradation during PINN model training.

## 3 ANALYSIS OF THE LEARNING STABILITY

**The effect of balancing method.** Weighting high residual regions during PINN training significantly enhances model accuracy and efficiency by minimizing errors and accelerating convergence (Lu et al., 2021b, Li et al., 2022). Additionally, it ensures stability (Cai et al., 2021a, Wang et al., 2021) and maintains physical consistency (Karniadakis et al., 2021, Wang et al., 2022b, Tang et al., 2023). These studies, motivated by the goal of improving model accuracy and efficiency, have empirically demonstrated the benefits of balancing method. However, there is a notable lack of theoretical analysis regarding the concentration of sampling in high residual regions. In this section, we aim to investigate the impact of sampling concentration through the resulting analysis.

**Setup.** Consider the partial differential equation defined over the domain $\Omega = \mathcal{X} \times [0, T]$. Assume that we have $N$ collocation points forming the population $\mathcal{P} = \{\mathbf{x}_n \in \Omega\}_{n=1}^N$, sampled from a uniform distribution $\mathcal{U}(\Omega)$.

**Assumption 3.1** *For analytical simplicity, we assume that the residual error of the PDE at each collocation point $\mathbf{x}_n$ can be expressed as a linear combination of feature-mapped vectors, given an appropriate feature map $\phi : \Omega \to \mathbb{R}^D$. Specifically, we represent the residual error as follows:*

$$\mathcal{R}_\theta(\mathbf{x}_n) = \frac{\partial}{\partial t} f_\theta(\mathbf{x}_n) + \mathcal{N}_x[f_\theta](\mathbf{x}_n) = a(\theta)^\intercal \phi(\mathbf{x}_n; \theta) = \sum_{d=1}^D a_d(\theta)\phi_d(\mathbf{x}_n; \theta) \qquad (3.1)$$

We regard the sampling methodology as a weighting of each sample point depending on the residual $\mathcal{R}_\theta(\mathbf{x}_n)$ and set $k = 2$. Thus, we can represent the loss function $\mathcal{L}(\mathcal{P}; \theta) = \sum_{n=1}^N w_n |\mathcal{R}_\theta(\mathbf{x}_n)|^2$. Assume that we are solving for the solution based on the gradient descent (GD) algorithm. Then,

$$\theta^{l+1} = \theta^l - \eta \nabla_\theta \left( \sum_{n=1}^N w_n^l |\mathcal{R}_{\theta^l}(\mathbf{x}_n)|^2 \right) \tag{3.2}$$

where the weights assigned to each sample point for iteration $l$ are determined as follows:

$$w_n^l \propto \exp\left( \frac{|\mathcal{R}_{\tilde{\theta}^l}(\mathbf{x}_n)|^2}{2\beta^2} \right), n \in \{1, ..., N\} \tag{3.3}$$

Additionally, $w_n^l$ is normalized to satisfy $\sum_{\mathbf{x} \in \mathcal{P}} w_n^l(\mathbf{x}) = 1$ and the parameter $\beta > 0$ preceding the residual controls the concentration of sampling with respect to the residuals. Note that the parameters $\tilde{\theta}^l = (\tilde{\theta}_1^l, \ldots, \tilde{\theta}_D^l)$ used to calculate the importance weights do not participate in the model parameter update process. Furthermore, in contexts where the meaning is clear, we will no longer explicitly indicate that $\phi$ is parameterized by $\theta$, i.e., denote $\phi(\mathbf{x}; \theta)$ as $\phi(\mathbf{x})$.

For iteration $l$, we focus on two extreme cases of interest: when $\beta$ is too large (uniform sampling), most samples receive uniform weights, resulting in uniform sampling. Conversely, when $\beta$ is close to 0 (high residual sampling), the effect is dominated by the sample with the highest residual. To explore this in more depth, consider the following propositions.

**Proposition 3.1 (Uniform sampling eigenvalue)** *When the sampling concentration parameter $\beta$ is sufficiently large, the maximum eigenvalue of the hessian of the loss function can be approximated as $2\lambda_{\max}(\Sigma)$, where $\Sigma = \mathbb{E}_{\mathbf{x} \sim \mathcal{U}(\Omega)}[\phi(\mathbf{x})\phi(\mathbf{x})^\intercal]$ and $\lambda_{\max}(\Sigma)$ is the maximum eigenvalue of $\Sigma$.*

**Proposition 3.2 (High residual sampling eigenvalue)** *When the sampling concentration parameter $\beta$ is sufficiently small, the maximum eigenvalue of the hessian of the loss function can be approximated as $2\|\phi(\mathbf{x}^*)\|^2$, where $\mathbf{x}^* = \arg\max_{\mathbf{x} \in \mathcal{P}} |\mathcal{R}_\theta(\mathbf{x})|^2$.*

Detailed proof can be found in Appendix B.1, B.2. It is well known that to ensure the convergence of GD algorithms, the learning rate $\eta$ must satisfy the following relationship with the largest eigenvalue $\lambda_{\max}$ of the hessian of the loss function: $\eta < \frac{2}{\lambda_{\max}}$ (Boyd & Vandenberghe, 2004). Therefore, we consequently aim to examine the relationship of the largest eigenvalue in two extreme cases of $\beta$. Before proceeding with the main result, we would like to introduce two assumptions.

**Assumption 3.2** *In high-dimensional feature space, $\|\phi(\mathbf{x})\|$ follows a heavy-tailed distribution. More specifically, $\mathbb{P}(\|\phi(\mathbf{x})\| > \zeta) \sim \frac{g(\zeta)}{\zeta^\alpha}$ for large $\zeta$, where $\sim$ represents asymptotic equivalence, $g(\zeta)$ satisfies $\forall t > 0, \lim_{\zeta \to \infty} \frac{g(t\zeta)}{g(\zeta)} = 1$ and $\alpha > 0$ indicates the thickness of the tail.*

This assumption is substantiated by both empirical evidence and theoretical insights. The heavy-tailed nature of feature vectors has been documented in several research results (Mahoney & Martin, 2019, Martin & Mahoney, 2020, Barsbey et al., 2021) and is theoretically supported by extreme value theory (Beirlant et al., 2006, Haan & Ferreira, 2006, Resnick, 2007) and random matrix theory (Pastur & Shcherbina, 2011, Tao, 2012).

**Assumption 3.3** *For the residual maximal point $\mathbf{x}^* = \arg\max_{\mathbf{x} \in \mathcal{P}} |\mathcal{R}_\theta(\mathbf{x})|^2$, as the dimension $D$ increases, it holds that $\left(\max_{\mathbf{x} \in \mathcal{P}} \|\phi(\mathbf{x})\|\right)^2 - \lambda_{max}(\Sigma) \gg \left(\max_{\mathbf{x} \in \mathcal{P}} \|\phi(\mathbf{x})\|\right)^2 - \|\phi(\mathbf{x}^*)\|^2$.*

This assumption can be seen as a weaker form of the concentration of measure phenomenon in high-dimensional spaces (Dubhashi & Panconesi, 2009, Vershynin, 2018, Nadjahi et al., 2021, Gupta et al., 2023). As dimensionality increases, random vectors concentrate around a typical norm, making the maximal norm representative of all vector norms. We assume that the gap between high residual points and the maximal norm changes slowly relative to $\lambda_{\max}(\Sigma)$. These assumptions were experimentally validated, as detailed in Appendix A. Assuming the number of samples $N$ scales with model size $D$ as $N = cD$, the following theorem can be derived.

**Theorem 3.1** *Given the heavy-tailed nature of $\|\phi(\mathbf{x})\|$ and sufficiently large model complexity $D$, we have $2\|\phi(\mathbf{x}^*)\|^2 \gg 2\lambda_{\max}(\Sigma)$. This inequality establishes a tighter upper bound on the learning rate for ensuring the convergence of the GD algorithm under the high residual sampling method.*

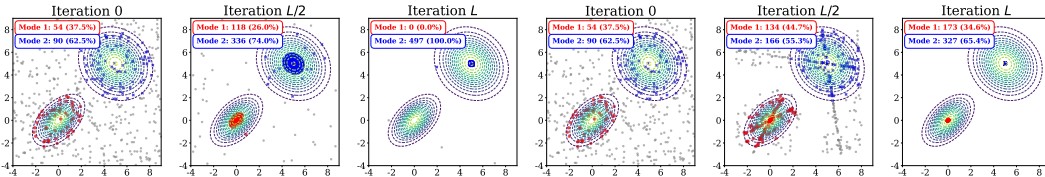

Figure 1: Schematic sampling diagram of (a) R3 (first three figures), (b) $L^\infty$ (last three figures) where $|\mathcal{R}_\theta(\mathbf{x})|^k = 0.3 \times \mathcal{N}\left(\mathbf{x}; \mathbf{0}, \left[[1, 0.5], [0.5, 1]\right]\right) + 0.7 \times \mathcal{N}\left(\mathbf{x}; \mathbf{5}, \left[[2, -0.3], [-0.3, 2]\right]\right)$.

The detailed proof can be found in the Appendix B.3. This indicates that the stability of the algorithm can vary significantly depending on the sampling method and model complexity. Specifically, during the actual training process with a weight decay scheme, learning may not progress adequately until the learning rate is sufficiently reduced, potentially compromising stability. Consequently, in these two extreme cases, uniform sampling may struggle to find an appropriate solution due to the complexity of the PDE problems, while high residual sampling may fail due to instability in the learning process. In this context, the unresolved issues can be summarized as follows.

**Limitations of prior works:**

1. **Imperfectness of sampling algorithms.** Most balancing sampling methods have not been precisely implemented when $\beta$ is at a moderate value. It is important to clarify that a moderate value of $\beta$ implies the ability to accurately describe the distribution proportional to the residual value. In particular, there has been insufficient consideration of the trajectory of samples used by algorithms to effectively update the PINN model, especially in the context of multi-modal residual landscapes with various scales of peaks.

   - **RAD** (Wu et al., 2023): The modeling of residual distribution is relatively straightforward and relies on monte carlo integration (MCI) over the expectation $\mathbb{E}|\mathcal{R}_\theta(\mathbf{x})|^k \approx \frac{1}{N}\sum_n |\mathcal{R}_\theta(\mathbf{x}_n)|^k$, which can be dependent on the number of sample points.
   - **R3** (Daw et al., 2023): R3 employs a strategy that consistently maintains high residuals, leading to an excessive skew in the distribution of collocation points. Moreover, this approach fails to effectively handle multi-modal landscapes in the long-term, which, as demonstrated in our previous theoretical analysis, results in a scenario where the sampling concentration parameter $\beta$ becomes extremely small.
   - **$L^\infty$** (Wang et al., 2022a): During the adversarial training, to estimate the inner maximal value $\sup_{\mathbf{x} \in \Omega} |\mathcal{R}_\theta(\mathbf{x})|^k$, $L^\infty$ iteratively utilizes gradient information $\text{sign}\nabla_\mathbf{x} |\mathcal{R}_\theta(\mathbf{x})|^k$ in the residual landscape with respect to $\mathbf{x}$, allowing for some degree of access to local modes. However, there is no guarantee that the proportions of modes with different heights will be maintained.

   To facilitate a more intuitive understanding of time evolving sampling methods (R3, $L^\infty$), we have illustrated the working mechanisms in a schematic diagram shown in Figure 1.

2. **Scalability with respect to model complexity.** Previous studies have primarily assessed algorithm effectiveness using small-scale model architectures. Consequently, even with relatively small values of $\beta$ (high residual sampling), these algorithms avoided instability during training and benefitted from the concentration effect that aids convergence. However, this limited evaluation raises concerns about their applicability in real-world scenarios. In this regard, Wang et al., 2024 made a notable contribution by proposing an architecture and initialization strategy designed to enhance stability across model complexities. In contrast, this study addresses the issue from the perspective of an adaptive sampling strategy.

## 4 PROPOSED APPROACH: LANGEVIN PINN (L-PINN)

Similar to other residual distribution-based methodologies, our primary objective is to estimate the residual-based sampling distribution. However, unlike previous methods that directly model the distribution using residuals, we employ Langevin dynamics to model the target distribution. An intuitive visualization of our L-PINN framework is depicted in Figure 2.

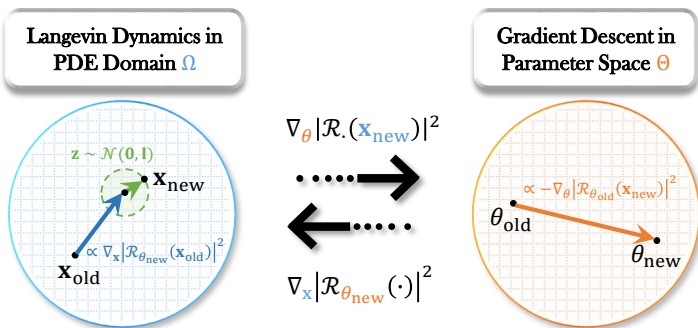

Figure 2: Bidirectional update: (Left) sample update in PDE domain, our L-PINN framework leverages Langevin dynamics to adaptively update collocation points based on PDE residuals at each iteration while keeping the PINN model $f_\theta$ fixed. (Right) parameter update in parameter space, conversely, the PINN model $f_\theta$ minimizes the PDE residuals with the updated collocation points. This iterative procedure continues to refine the solution until convergence is achieved.

## 4.1 LANGEVIN DYNAMICS AND STATIONARY DISTRIBURION

The dynamics of the collocation points at the $l$-th iteration utilized in Langevin PINN (L-PINN) can be described as follows:

$$\mathbf{x}_n^{l+1} = \mathbf{x}_n^l + \frac{\tau}{2}\nabla_{\mathbf{x}}|\mathcal{R}_\theta(\mathbf{x}_n^l)|^2 + \beta\sqrt{\tau}\mathbf{z}_n^l, n \in \{1, ..., N\} \tag{4.1}$$

where $\tau > 0$ is the step size, $\mathbf{z}_n^l \sim \mathcal{N}(\mathbf{z}_n^l; \mathbf{0}, \mathbf{I})$ represents the white Gaussian noise, and $\beta$ is the sampling concentration coefficient. Additionally, the residual exponent $k$ is set to 2. Unlike other methods that estimate the sampling distribution based on residuals at every iteration, L-PINN dynamically updates the data points without requiring the estimation of the sampling distribution. If such Langevin dynamics are allowed to run for a *sufficient number of iterations* with *sufficiently small step size*, we can theoretically obtain the following result for the collocation points.

**Theorem 4.1 (Stationary distribution)** *For fixed $f_\theta$ and concentration parameter $\beta > 0$, sample population $\mathcal{P}^l$ asymptotically follows* $\lim_{l\to\infty} p_l(\mathbf{x}) = p(\mathbf{x}) \propto \exp\left(\frac{|\mathcal{R}_\theta(\mathbf{x})|^2}{2\beta^2}\right)$ *as $l \to \infty$.*

The proof of Theorem 4.1 can be found in the Appendix C.1. As evident from the above results, the L-PINN framework can achieve collocation sample population at an arbitrary $\beta > 0$. This differs from methods like R3 and RAD, which sample new collocation points multiple times to find high residuals. The L-PINN conducts successive sampling by using the evolving population as the initial point for the next update. It can also be compared to $L^\infty$, which fully initializes the collocation points at each iteration and leverages the gradient information as $\text{sign}\nabla_{\mathbf{x}}|\mathcal{R}_\theta(\mathbf{x})|^k$ multiple times to identify the local mode, ensuring that the directional information of the gradient vectors is preserved. The detailed operational procedure can be found in Algorithm 1.

---

**Algorithm 1** Single L-PINN Sampling Iteration for Physics-Informed Neural Networks

---

1: **Input: initial population $\mathcal{P} = \mathcal{P}^0$ with $N$ collocation points**
2: **Output: updated population $\mathcal{P} = \mathcal{P}^{l_\mathrm{L}}$**
3: **for** $l = 0$ to $l_\mathrm{L} - 1$ **do**
4:     **for** $\mathbf{x}_n^l \in \mathcal{P}^l$ **do**
5:         **Calculate the gradient:** $\nabla_{\mathbf{x}}|\mathcal{R}_\theta(\mathbf{x}_n^l)|^2 = \nabla_{\mathbf{x}}\left|\frac{\partial}{\partial t}f_\theta(\mathbf{x}_n^l) + \mathcal{N}_x[f_\theta(\mathbf{x}_n^l)]\right|^2$
6:         **Sample white Gaussian noise:** $\mathbf{z}_n^l \sim \mathcal{N}(\mathbf{z}_n^l; \mathbf{0}, \mathbf{I})$
7:         **Follow the Langevin dynamics:** $\mathbf{x}_n^{l+1} \leftarrow \mathbf{x}_n^l + \frac{\tau}{2}\nabla_{\mathbf{x}}|\mathcal{R}_\theta(\mathbf{x}_n^l)|^2 + \beta\sqrt{\tau}\mathbf{z}_n^l$
8:     **end for**
9:     **Update collocation population:** $\mathcal{P}^{l+1} \leftarrow \{\mathbf{x}_n^{l+1}\}_{n=1}^N$
10: **end for**

---

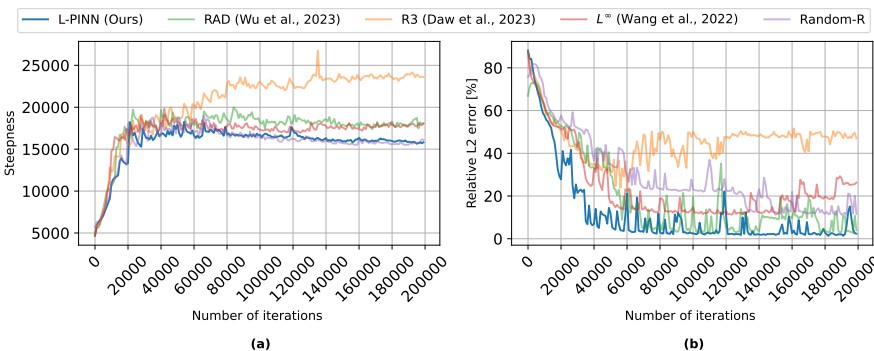

Figure 3: With fixed learning rate $\eta = 0.002$ and 4 hidden layers, (a) The maximal eigenvalue of the hessian (steepness) for the loss function, (b) the relative $L^2$ error curve.

## 4.2 PRACTICAL IMPLEMENTATION

In general, achieving successful Langevin sampling requires careful selection of hyperparameters (step size $\tau$, number of Langevin iterations $l_{\mathrm{L}}$, etc.). While running Langevin dynamics for many iterations with a small step size might allow sampling to be proportional to the actual residual landscape, it can significantly slow down the training speed of the PINN model in practical applications. Thus, we considered the following concepts when setting the hyperparameters, which are crucial for effectively utilizing Langevin dynamics.

**Adjusting step size and Langevin iteration.** To increase the computational efficiency, we adopted a strategy of increasing the step size $\tau$ and reducing the number of Langevin iterations $l_{\mathrm{L}}$, even at the cost of some loss in sample quality. We anticipated that the temporal variation of the PINN model, $f_\theta$, would exhibit smooth behavior when utilizing adaptive sampling strategies. Consequently, even with fewer Langevin iterations, minimal changes in the loss landscape suggest that the sample trajectory would resemble that of a fixed landscape.

**Normalizing the gradient size.** Since L-PINN leverages gradient information from the residual landscape, the step size $\tau$ needs to be set even smaller for stiff PDEs. Additionally, empirical observations indicate that the gradient of the residual landscape exhibits substantial variations at the beginning of training. In contrast, towards the end of training, the residual landscape is characterized by relatively small gradients. This discrepancy restricts the movement of sample points, thereby making it challenging to secure reliable quality. To address these challenges, we normalized the magnitude of all residual gradients at each iteration relative to the largest residual gradient. This method effectively mitigated the hyperparameter sensitivity inherent to Langevin dynamics.

## 5 EXPERIMENTS

In this section, we experimentally evaluate the effects of focusing on high residuals, considering variations in model complexity while keeping the number of collocation points fixed (noting that, in general, model performance improves as $N_{\mathrm{pde}}$ increases). We compare the performance of our L-PINN against other adaptive sampling methods, including RAD, R3, $L^\infty$, and Random-R which uniformly resamples all collocation points at each iteration.

**Experimental setup.** As the default settings, unless otherwise specified, the models utilized a multilayer perceptron (MLP) with 128 nodes per layer and 4 hidden layers, employing a hyperbolic tangent activation function in each hidden layer. The Adam optimizer was utilized with the learning rate of $\eta = 0.001$ and a decay factor of 0.9 applied every 5,000 iterations. Training was conducted with 200,000 iterations, and the number of collocation points was set to $N_{\mathrm{pde}} = 1,000$. For the L-PINN configuration, the residual exponent $k = 2$, the Langevin step size $\tau = 0.002$, and the concentration parameter $\beta = 0.2$.

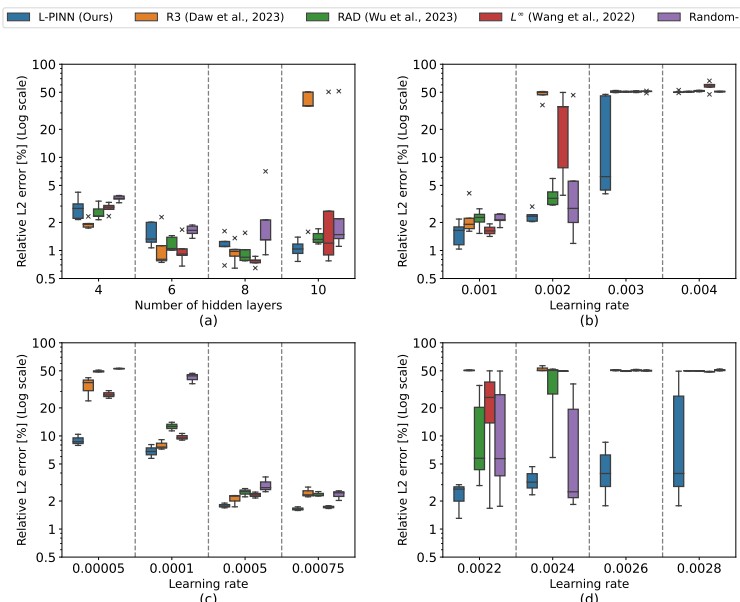

Figure 4: Relative $L^2$ error (Log scale) for the $\mathrm{Allen-Cahn}$ equation: (a) varying layers with $\eta = 0.001$ (with scheduler), (b)-(d) fixed 4 layers with different learning rates (no scheduler). Each boxplot is based on 5 random seeds.

## 5.1 ABLATION STUDIES

First and foremost, we sought to verify how the analytical results regarding stability and model complexity, presented in Section 3, operate and apply to the functioning of each algorithm. In this context, we performed the following key ablation studies based on the $\mathrm{Allen-Cahn}$ equation using 5 different random seeds. To validate the behaviors discussed in Section 4.2, the sample trajectory recorded during the actual training process is provided in Appendix D.

**Steepness of the loss landscape.** In our stability analysis, we posited that the loss landscape of a sampling algorithm targeting extremely high residuals, such as R3, $L^\infty$ would exhibit sharp landscapes. To validate this, we visualized the maximal eigenvalue of the hessian within the loss across iterations in Figure 3-(a). The results confirm our hypothesis that sampling methods focused on extreme high residuals lead to greater steepness. However, as shown in Figure 3-(b), despite the stability of Random-R, its modest performance suggests that low steepness alone does not ensure better sample quality or performance, emphasizing the need for concentration of high residual points.

**Different number of hidden layers.** We employed MLP architectures with hidden layers ranging from 4 to 10 across all sampling methods, maintaining a learning rate of 0.001 and utilizing a step scheduler. As illustrated in Figure 4-(a), it can be observed that only L-PINN and RAD demonstrated stable performance when 10 hidden layers were used. These results indicate that high residual methods are more susceptible to increasing model complexity, whereas L-PINN remains robust. Under various settings, the detailed experimental results are presented in Appendix E.

**Varying learning rate $\eta$ without decaying.** We evaluated MLPs with four hidden layers across learning rates ranging from 0.001 to 0.004 without applying decay. As shown in Figure 4 -(b), the benchmark algorithms demonstrated performance degradation at $\eta = 0.002$ compared to $\eta = 0.001$, whereas L-PINN showed improvement. At $\eta = 0.003$, all methods exhibited reduced performance; however, L-PINN was able to partially mitigate this degradation. At $\eta = 0.004$, none of the methods produced correct solutions. In particular, we visualized the performance for very low learning rates in Figure 4-(c) and highlighted the range between $\eta = 0.002$ and 0.003, where all algorithms begin to exhibit instability in Figure 4-(d). From this, we observe that learning does not proceed properly at very low learning rates, and for layer 4, most algorithms become unstable at a learning rate as low as approximately 0.0022.

## 5.2 ADDITIONAL EXPERIMENTS ON REPRESENTATIVE PDEs

The proposed L-PINN framework is further evaluated on representative 1D PDEs derived from various benchmark problems tackled by several established algorithms, including RAD, R3, $L^\infty$, Random-R. In these evaluations, we employ the default experimental settings as outlined earlier. The specific configurations for the PDE parameters and the hyperparameters of the baseline algorithms are detailed in Appendix F. While fine-tuning hyperparameters like Langevin iterations, concentration parameter, and step size may improve performance, our main goal is to showcase the robustness of the L-PINN framework to these hyperparameters. Additional results on hyperparameter sensitivity, computational complexity, PDE dimensionality effects, and compatibility across architectures are provided in Appendices G, H, I, and J, respectively.

**Experimental results.** We evaluated the performance of each sampling method on the Burgers′, Convection, Allen−Cahn, Korteweg−DeVries, and Schrödinger equations across 5 different random seeds. The results in Table 1 indicate that L-PINN generally achieves superior or comparable relative $L^2$ errors compared to other models. For the Burgers′ equation, Random-R performed best, with L-PINN close behind. In the Convection equation, RAD outperformed others, while $L^\infty$ failed to converge correctly. For the Allen−Cahn equation, L-PINN achieved the best performance, followed by $L^\infty$. Results for layer 10 are shown in Fig. 5. In the Korteweg−DeVries equation, Random-R ranked first, and L-PINN ranked second, with other methods producing larger errors. For the Schrödinger equation, L-PINN performed best, followed by Random-R. Overall, L-PINN and Random-R consistently demonstrated superior performance across PDEs.

Table 1: Relative $L^2$ error across PDEs for increasing model complexity with larger hidden layers.

| PDEs | Burgers′ | | Convection | | Allen−Cahn | | Korteweg−DeVries | | Schrödinger | |
|---|---|---|---|---|---|---|---|---|---|---|
| Number of layers | 8 | 10 | 8 | 10 | 8 | 10 | 8 | 10 | 8 | 10 |
| Random-R | **0.01±0.00** | 0.02±0.00 | 0.30±0.05 | 0.41±0.10 | 2.54±2.30 | 11.49±19.93 | **1.64±0.63** | 2.89±1.80 | 0.09±0.00 | 0.11±0.01 |
| RAD | 0.17±0.02 | 0.27±0.14 | **0.25±0.02** | 0.28±0.09 | 0.99±0.29 | 1.36±0.19 | 7.44±1.83 | 7.97±1.45 | 1.68±0.15 | 2.89±0.69 |
| R3 | **0.01±0.00** | 0.02±0.00 | 0.39±0.24 | 0.27±0.05 | 0.97±0.23 | 34.47±17.64 | 3.92±2.93 | 7.02±8.77 | 0.11±0.01 | 0.15±0.02 |
| $L^\infty$ | 0.03±0.01 | 0.06±0.06 | 73.87±5.07 | 54.17±27.33 | 0.76±0.07 | 10.95±19.16 | 5.70±1.45 | 4.44±1.45 | 0.22±0.06 | 0.19±0.03 |
| L-PINN (ours) | **0.01±0.00** | **0.01±0.00** | 0.34±0.12 | **0.27±0.03** | **0.75±0.11** | **1.06±0.21** | 2.68±1.74 | **1.99±0.50** | **0.08±0.01** | **0.09±0.01** |

## 6 CONCLUSIONS

In this paper, we analyzed the impact of adaptive sampling methods on learning stability when training PINN models, particularly in relation to model complexity. Our theoretical analysis revealed that sampling methods overly focused on high residuals could compromise learning stability. To mitigate this issue, we proposed the Langevin dynamics-based PINN (L-PINN) framework, which updates collocation points based on Langevin dynamics proportional to PDE residuals. Through extensive experiments and ablation studies, we demonstrated that high residual-based methods often failed to converge to correct solutions when increasing hidden layers and learning rate rates, whereas L-PINN maintained stable convergence.

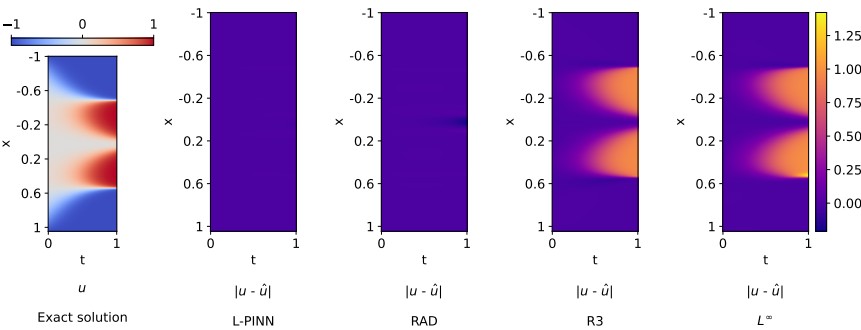

Figure 5: Error comparison of the exact solution and the predicted value for the Allen−Cahn equation at layer 10 across benchmark algorithms.

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

# APPENDIX

## A EMPIRICAL VALIDATION OF ASSUMPTIONS

We begin by discussing the challenges of validation associated with our assumption, which states that proving a linear combination of the residual function to its feature vector is non-trivial. By definition, the residual $\mathcal{R}_\theta$ involves transforming the neural network output $f_\theta$ through an operator within a certain function space. This implies that the features of the function, resulting from the neural network output combined with an additional operator, are not explicitly defined. Consequently, we extract the feature vector of the resulting function through local feature vector estimation based on the linearization of the residual function. Before delving into the main discussion, we first explain the logic behind how the feature vector $\phi$ is inferred.

### A.1 LOCAL APPROXIMATION OF THE FEATURE VECTOR $\phi$

Let $\mathcal{R}_\theta(\mathbf{x}) = \frac{\partial}{\partial t} f_\theta(\mathbf{x}) + \mathcal{N}_x[f_\theta](\mathbf{x})$ represent $[g(f_\theta)](\mathbf{x})$. To validate the assumption that a suitable linearization exists, our goal is to derive a proper linear approximation of $[g(f_\theta)](\mathbf{x})$ at a specific point $\mathbf{x} \in \Omega = \mathcal{X} \times [0, T]$, given a specific function $f_\theta$. In this process, we will utilize a Taylor expansion for $[g(f_\theta)](\mathbf{x})$. It is important to note that since $g(f_\theta)$ represents the behavior in the function space, understanding how $g$ responds to small perturbations in $f_\theta$ is crucial. This analysis employs the Fréchet derivative.

To summarize briefly, $g(f_\theta) \approx g(f) + D_g(f)(f_\theta - f)$, which implies that the result can be linearized around a baseline function $f$ where $D_g(f) = \lim_{\Delta f \to 0} \frac{||g(f + \Delta f) - g(f)||}{||\Delta f||}$. Since our focus is on the linearization of $[g(f_\theta)](\mathbf{x})$, it is essential to ensure that $f$ is a function close to $f_\theta$ within the function space. To achieve this, small noise perturbations are added to the neural network $f_\theta$. In conclusion, to approximate the value at a specific point $\mathbf{x}$, we proceed as follows:

$$[g(f_\theta)](\mathbf{x}) \approx [g(f) + D_g(f)(f_\theta - f)](\mathbf{x}) \tag{A.1.1}$$

$$= [g(f)](\mathbf{x}) + [D_g(f)(f_\theta - f)](\mathbf{x}) \tag{A.1.2}$$

$$= [g(f)](\mathbf{x}) + [D_g(f)](\mathbf{x})\big(f_\theta(\mathbf{x}) - f(\mathbf{x})\big) \tag{A.1.3}$$

Here, if $f_\theta$ is assumed to be a well-trained PINN model and perturbation $\Delta f$ is sufficiently small, we can readily infer the following for the first term:

$$[g(f)](\mathbf{x}) \approx 0, \quad \forall \mathbf{x} \in \Omega.$$

Consequently, the linear approximation of the function $[g(f_\theta)](\mathbf{x})$ can be expressed using the Fréchet derivative. The aspect that conflicts with our assumption is that, in this context, the Fréchet derivative can act as a function dependent on $\mathbf{x}$. Therefore, we refer to this as a local approximation.

**Approximation of Fréchet derivative.** According to the problem formulation of PINN, $g$ is an operator that takes the function $f$ as input and generates new values through partial derivatives such as $f_x, f_t, f_{xx}, f_{xt}$, and their combinations. Thus, we can assume $g(f) = G(f, f_x, f_t, f_{xx}, f_{xt}, \cdots)$. Here, $G$ is a multivariate function that combines the derivative terms. Next, considering a scenario where a slight perturbation $\Delta f$ is applied to $f$, the Fréchet derivative can be approximated as follows:

$$D_g(f)(f_\theta - f) \approx g(f + \Delta f) - g(f) \tag{A.1.4}$$

$$= G(f + \Delta f, f_x + \Delta f_x, f_t + \Delta f_t, \cdots) - G(f, f_x, f_t, \cdots) \tag{A.1.5}$$

$$= \frac{\partial G}{\partial f} \Delta f + \frac{\partial G}{\partial f_x} \Delta f_x + \frac{\partial G}{\partial f_t} \Delta f_t + \cdots \tag{A.1.6}$$

$$\approx \sum_k \frac{\partial G}{\partial f_k} \Delta f_k, \quad f_k \in \{f, f_x, f_t, f_{xt}, \cdots\} \tag{A.1.7}$$

$$=: a^\mathsf{T} \phi \tag{A.1.8}$$

where $a = \left(\frac{\partial G}{\partial f}, \frac{\partial G}{\partial f_x}, \frac{\partial G}{\partial f_t}, \cdots\right)$ and $\phi = (\Delta f, \Delta f_x, \Delta f_t, \cdots)$.

**Mathematical details.** Here, we provide a systematic summary of the considerations underlying the validity of the employed estimation method.

1. **Reliability of the Fréchet derivative** $D_g[f]$**:** The existence of the Fréchet derivative requires the following sufficient conditions:

   - $G$ must be differentiable.
   - $f$ must be sufficiently differentiable with respect to $x$ and $t$.

   Both conditions are naturally satisfied in the context of our problem. This ensures that we can extract a vector that locally approximates the actual feature vector $\phi$ for each point $\mathbf{x} \in \Omega$, thereby facilitating a robust estimation process.

2. **Condition for the constancy of** $D_g[f]$**:** It is important to note that $G$ is generally a function of $(f, f_x, f_t, \cdots)$, and thus implicitly depends on $\mathbf{x}$. However, when the variables are not entangled with each other, the partial derivatives can exhibit constant behavior. For instance:

   - In our case, $\frac{\partial G}{\partial f_t}$ is always 1.
   - Partial derivatives in the $x$-direction such as $\frac{\partial G}{\partial f}, \frac{\partial G}{\partial f_x}, \frac{\partial G}{\partial f_{xx}}$ depend on $\mathcal{N}_x$.

   If the output of the differential operator $\mathcal{N}_x$ entangles the partial derivatives in the $x$-direction (i.e, $\mathcal{N}_x[f]$ is non-linear), the assumption that $a$ acts as a constant may weaken.

Now, to compute the quantity $\phi$ defined in this manner, we use $\Delta f_x \approx \frac{\partial f_x}{\partial x}\Delta x + \frac{\partial f_x}{\partial t}\Delta t$, and the other $\Delta f_\circ$ values for the remaining partial derivatives can be computed similarly. Furthermore, partial derivatives of the neural network $f$ with respect to $(x, t)$ can be approximated using automatic differentiation.

## A.2 Heavy-Tailed Behavior of the Norm of Feature Vectors

Initially, we visualized the histogram of the norms of the extracted feature vectors across all feasible grid points in Figure 6, i.e., the histogram of $\{\|\phi(\mathbf{x})\| : \mathbf{x} \in \Omega = \mathcal{X} \times [0, T]\}$ for models with 4, 6, 8, and 10 layers, respectively.

From the provided histograms, it is evident that for each PDE, the distribution increasingly exhibits heavy-tail behavior as the layer depth grows. This tendency is particularly emphasized in the following two aspects:

1. **Heavy-tail characteristics resembling Pareto distribution:** As the layer depth increases, the distribution's tail becomes thicker, consistent with the heavy-tail properties of the Pareto distribution. In a Pareto distribution, the tail probability follows the form $\mathbb{P}(X > x) \propto x^{-\alpha}$, decaying slowly and exhibiting a high frequency of extreme values. This is reflected in the histograms, where deeper layers show data concentrated in certain regions while displaying more frequent extreme values.

2. **Increased concentration and frequency of extreme values:** As the number of layers increases, the data become densely concentrated within specific ranges (represented on the $y$-axis as frequency), while significantly more frequent occurrences of large values (depicted on the $x$-axis as extreme values) are observed. This behavior suggests a progressive shift towards heavy-tail distributions.

In addition to the previously obtained histograms, we also calculated two statistical estimates—Pareto tail index and Hill estimator—based on the samples to provide a more quantitative representation.

**Pareto tail index.** The Pareto tail index, denoted by $\alpha$, quantifies the heaviness of the tail of a distribution. For a random variable $X$ with a heavy-tailed distribution, the tail probability follows a power-law:

$$\mathbb{P}(X > x) \sim x^{-\alpha}, \quad \text{as } x \to \infty,$$

where $\alpha > 0$ represents the tail index. Therefore, a smaller value of $\alpha$ corresponds to a thicker tail, indicating a slower decay of the tail probability and a higher likelihood of extreme events.

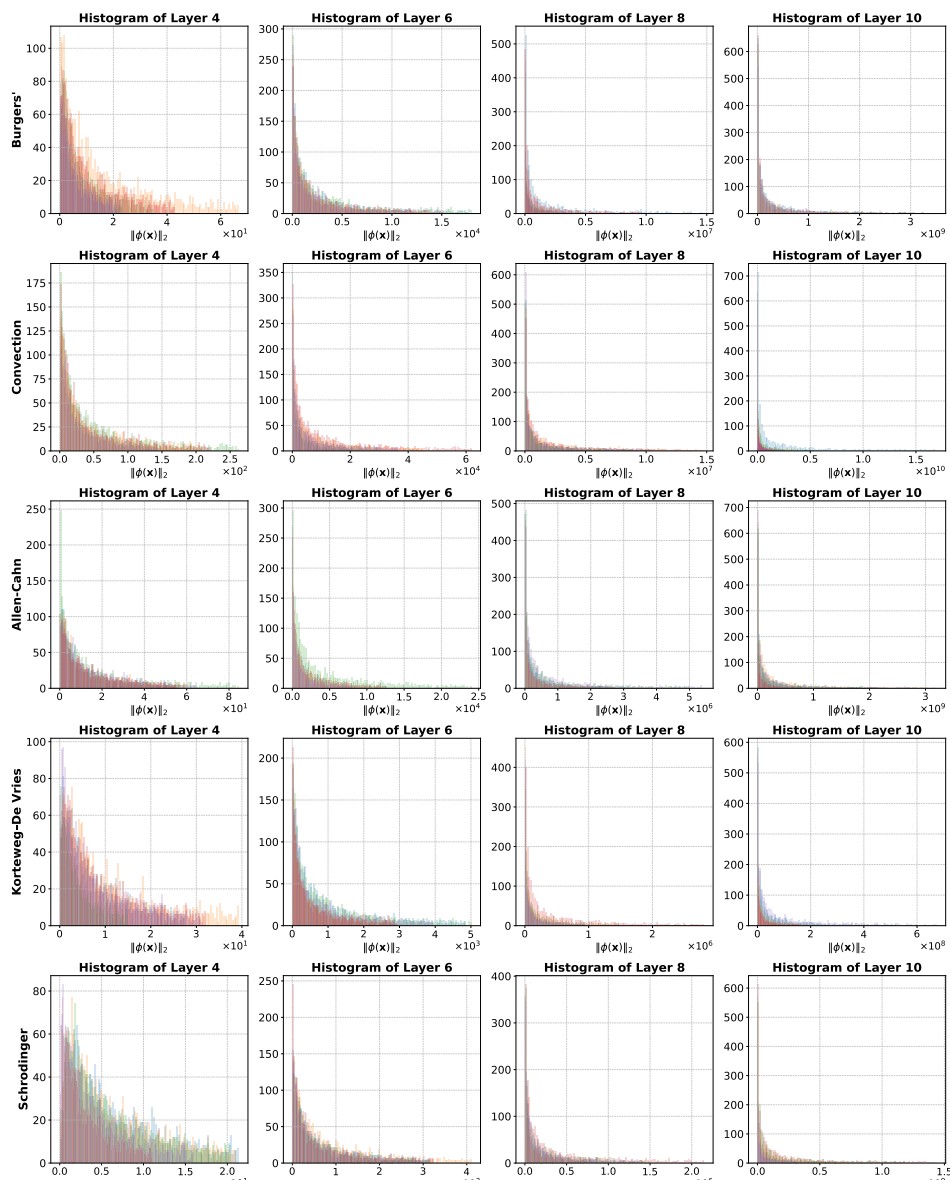

Figure 6: Each plot represents a (PDE, layer) pair, where the row corresponds to the type of PDE being solved (e.g., Burgers′, Convection, Allen−Cahn, etc.), and the column indicates the model size by the number of hidden layers in the PINN (e.g., layer 4, 6, 8, 10). The histograms show the distributions of the feature vector norms $\|\phi(\mathbf{x})\|$ for each pair.

Conversely, a larger value of $\alpha$ corresponds to a thinner tail, where the tail probability decays more rapidly and extreme events are less likely.

**Hill estimator.** The Hill estimator is specifically designed to estimate the inverse of the tail index, $\xi = \frac{1}{\alpha}$. Given a sample of $n$ independent and identically distributed observations $\{X_1, X_2, \ldots, X_n\}$, sorted in descending order as $X_{(1)} \geq X_{(2)} \geq \cdots \geq X_{(n)}$, the Hill estimator is defined as:

$$\hat{\xi}_k = \frac{1}{k} \sum_{i=1}^{k} \log \frac{X_{(i)}}{X_{(k+1)}},$$

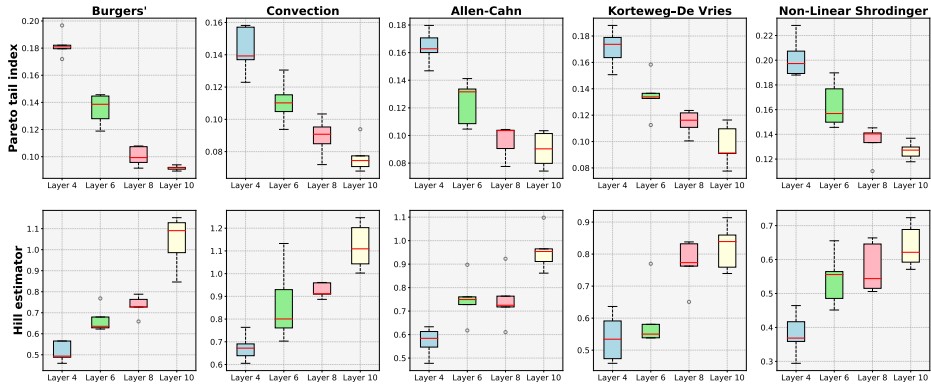

Figure 7: Two statistical estimates of the norms of the feature vectors.

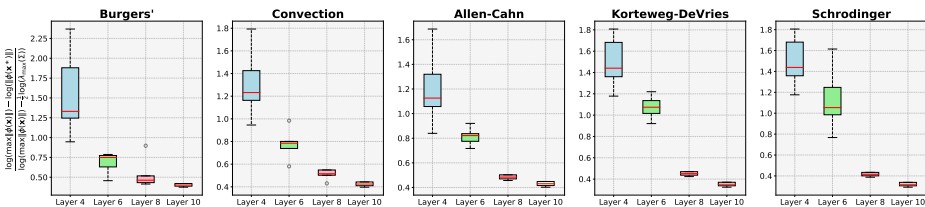

Figure 8: Disparity comparison of $\lambda_{\max}(\Sigma)$ and $\phi(\mathbf{x}^*)$ with respect to $\max \|\phi(\mathbf{x})\|$.

where $k$ is the number of upper order statistics used for the estimation. It is evident that $\hat{\xi}_k$ estimates the reciprocal of the true tail index $\alpha$. Consequently, a larger value of $\hat{\xi}_k$ corresponds to a smaller $\alpha$, which can be interpreted as indicating a heavier tail.

Figure 7 presents box plots of the estimates across different random seeds for various PDEs. The Pareto tail index, quantifying tail thickness, decreases with increasing layers, indicating heavier tails and a higher likelihood of extreme events. Across all PDEs, the index consistently declines from layer 4 to layer 10, highlighting the growing dominance of extreme values in deeper layers. Notably, the index values are significantly below 2, a common threshold for heavy-tail behavior. Even with fewer layers and 1,000 collocation points, feature vector norms exhibit pronounced heavy-tail distributions. The Hill estimator, measuring the inverse of tail heaviness, complements this, gradually increasing with layer depth and reinforcing the trend of heavier tails. Its values, exceeding the 0.5 threshold for heavy tails, become more pronounced with depth.

## A.3 EMERGING DISPARITIES WITH INCREASING MODEL COMPLEXITY

In the previous subsection, we conducted an empirical analysis of the distributional characteristics of feature vector norms. In this subsection, we aim to validate the hypothesis of the norm's emerging disparities with increasing model complexity. (Assumption 3.3) For clarity, this relation can be expressed mathematically as $\left(\max_{\mathbf{x}\in\mathcal{P}} \|\phi(\mathbf{x})\|\right)^2 - \lambda_{\max}(\Sigma) \gg \left(\max_{\mathbf{x}\in\mathcal{P}} \|\phi(\mathbf{x})\|\right)^2 - \|\phi(\mathbf{x}^*)\|^2$, where $\mathbf{x}^* = \arg\max_{\mathbf{x}\in\Omega} |\mathcal{R}_\theta(\mathbf{x})|^2$. Here, due to the dominant scale of $\max \|\phi(\mathbf{x})\|$, we transformed the values into a logarithmic scale to investigate the relationship between $\lambda_{\max}$ and $\|\phi(\mathbf{x}^*)\|$.

The Figure 8 illustrates the behavior of a logarithmic metric for various PDEs as the layer count increases. The x-axis represents the number of layers, shown as 4, 6, 8, and 10, while the y-axis represents a log-based value, denoted as $\frac{\log(\max \|\phi(\mathbf{x})\|) - \log \|\phi(\mathbf{x}^*)\|}{\log(\max \|\phi(\mathbf{x})\|) - \frac{1}{2}\log \lambda_{\max}(\Sigma)}$, which captures a ratio involving maximum values and scaled terms. Across all PDEs, the y-axis value decreases monotonically as the number of layers increases. This consistent decline in the log-metric across all PDEs suggests that as the layer count grows, the denominator in the ratio scales disproportionately compared to the numerator. This behavior indicates that the underlying system dynamics or representation becomes increasingly dominated by the factors represented in the denominator.

## B   Learning Rate Upper Bound Varying $\beta$

For the sake of simplicity, we will consider the situation at iteration $l$. Hence, in the forthcoming proof, we will omit the upper index related to iteration. i.e., denote $w_n^l$ as $w_n$.

### B.1   Proof of Proposition 3.1

When the sampling concentration coefficient $\beta$ is sufficiently large, the weights $w_n$ are approximately uniform ($w_n \approx \frac{1}{N}$). Thus, the hessian matrix $\mathbf{H}_\theta$ of the loss function with respect to $\theta$ can be approximated by:

$$\mathbf{H}_\theta(\mathcal{L}) = \mathbf{H}_\theta \left( \sum_{n=1}^{N} w_n |\mathcal{R}_\theta(\mathbf{x}_n)|^2 \right) \approx \mathbf{H}_\theta \left( \frac{1}{N} \sum_{n=1}^{N} |\mathcal{R}_\theta(\mathbf{x}_n)|^2 \right) \tag{B.1.1}$$

Since $\mathcal{R}_\theta(\mathbf{x}) = \theta^\mathsf{T} \phi(\mathbf{x})$ and $\mathbf{H}_\theta \left( |\theta^\mathsf{T} \phi(\mathbf{x}_n)|^2 \right) = 2\phi(\mathbf{x})\phi(\mathbf{x})^\mathsf{T}$, the hessian of $\mathcal{L}$ satisfies:

$$\mathbf{H}_\theta(\mathcal{L}) \approx \frac{1}{N} \sum_{n=1}^{N} \mathbf{H}_\theta \left( |\theta^\mathsf{T} \phi(\mathbf{x}_n)|^2 \right) = \frac{2}{N} \sum_{n=1}^{N} \phi(\mathbf{x}_n)\phi(\mathbf{x}_n)^\mathsf{T} \tag{B.1.2}$$

This matrix represents the sample covariance matrix of feature vector $\phi(\mathbf{x})$. Thus, for sufficiently large $N$, we can say the maximum eigenvalue of the hessian is approximately $2\lambda_{\max}(\Sigma)$.

### B.2   Proof of Proposition 3.2

When the sampling concentration coefficient $\beta$ is sufficiently small, for the sample $\mathbf{x}^*$ with the largest residual (i.e., $\mathbf{x}^* = \arg\max_{\mathbf{x} \in \mathcal{P}} |\mathcal{R}_\theta(\mathbf{x})|^2$), we can consider all other weights to be zero except for $\mathbf{x}^*$. Therefore, the hessian matrix of the loss function can be expressed as follows:

$$\mathbf{H}_\theta(\mathcal{L}) = \mathbf{H}_\theta \left( \sum_{n=1}^{N} w_n |\mathcal{R}_\theta(\mathbf{x}_n)|^2 \right) \approx \mathbf{H}_\theta \left( |\mathcal{R}_\theta(\mathbf{x}^*)|^2 \right) \tag{B.2.1}$$

The hessian of the loss function can be expressed as $2\phi(\mathbf{x}^*)\phi(\mathbf{x}^*)^\mathsf{T}$, which is rank-1 matrix. Given that the eigenvalue equation is defined as $Av = \lambda v$, where $A = 2\phi(\mathbf{x}^*)\phi(\mathbf{x}^*)^\mathsf{T}$ and $v = \phi(\mathbf{x}^*)$, it follows that $\phi(\mathbf{x}^*)$ is the eigenvector and $2\|\phi(\mathbf{x}^*)\|^2$ is the corresponding eigenvalue. Since $A$ is rank-1 matrix, the eigenvalue is uniquely determined.

### B.3   Proof of Theorem 3.1

We have assumed that the norm of $\phi$ follows a heavy-tailed distribution (Assumption 3.2). Then, according to extreme value theory, it is well-established that the maximum value obtained from $N$ samples scales as $N^{1/\alpha}$, i.e., $\max \|\phi(\mathbf{x})\| \sim N^{1/\alpha}$. Given that $N = cD$, where $N$ is proportional to $D$ due to high-dimensionality and sampling considerations, we approximate:

$$\max \|\phi(\mathbf{x})\| \sim (cD)^{1/\alpha} \sim D^{1/\alpha} \tag{B.3.1}$$

Subsequently, the matrix $\Sigma = \mathbb{E}[\phi(\mathbf{x})\phi(\mathbf{x})^\mathsf{T}]$ represents the covariance matrix of the feature mappings $\phi(\mathbf{x})$. In high-dimensional settings, the eigenvalues of such covariance matrices are known to follow specific distribution patterns as described by random matrix theory. In particular, the maximum eigenvalue of $\Sigma$, denoted $\lambda_{\max}(\Sigma)$, scales as $N^{2/\alpha-1}$. Consequently, $\lambda_{\max}(\Sigma) \sim N^{2/\alpha-1}$. Substituting $N = cD$ into the scaling relationship yields:

$$\lambda_{\max}(\Sigma) \sim (cD)^{2/\alpha-1} \sim D^{2/\alpha-1} \tag{B.3.2}$$

When $D$ is large, the term $(\max \|\phi(\mathbf{x})\|)^2$, which scales as $D^{2/\alpha}$, increases significantly faster than $\lambda_{\max}(\Sigma)$, which scales as $D^{2/\alpha-1}$. Consequently, as $D$ increases, $(\max_{\mathbf{x} \in \mathcal{P}} \|\phi(\mathbf{x})\|)^2 \gg \lambda_{\max}(\Sigma)$. Finally, combining this fact with Assumption 3, which states $(\max_{\mathbf{x} \in \mathcal{P}} \|\phi(\mathbf{x})\|)^2 - \lambda_{\max}(\Sigma) \gg (\max_{\mathbf{x} \in \mathcal{P}} \|\phi(\mathbf{x})\|)^2 - \|\phi(\mathbf{x}^*)\|^2$, we can conclude the proof.

# C    STATIONARY DISTRIBUTION OF LANGEVIN DYNAMICS

## C.1    PROOF OF THEOREM 4.1

Given the modified Langevin dynamics, for any $\mathbf{x}_n^l \in \mathcal{P}^l$,

$$\mathbf{x}_n^{l+1} = \mathbf{x}_n^l + \frac{\tau}{2}\nabla_{\mathbf{x}}\left|\mathcal{R}_\theta(\mathbf{x}_n^l)\right|^2 + \beta\sqrt{\tau}\mathbf{z}_n^l \tag{C.1.1}$$

The drift and diffusion terms from the modified Langevin dynamics are:

$$\mathbf{A}(\mathbf{x}) = \frac{\tau}{2}\nabla_{\mathbf{x}}\left|\mathcal{R}_\theta(\mathbf{x})\right|^2, \tag{C.1.2}$$

$$\mathbf{B}(\mathbf{x}) = \beta^2\tau\mathbf{I} \tag{C.1.3}$$

If we denote the probability density of $\mathbf{x} \in \Omega$ at time $l$ as $p_l(\mathbf{x})$, the Fokker-Planck equation describing the time evolution of the probability density is:

$$\frac{\partial p_l(\mathbf{x})}{\partial l} = -\nabla_{\mathbf{x}} \cdot (\mathbf{A}(\mathbf{x})p_l(\mathbf{x})) + \nabla_{\mathbf{x}} \cdot (\mathbf{B}(\mathbf{x})\nabla_{\mathbf{x}}p_l(\mathbf{x})) \tag{C.1.4}$$

where $\cdot : \Omega \times \Omega \to \mathbb{R}$ represents general dot product between two vectors. i.e., $u \cdot v = u^{\mathsf{T}}v$ and $\nabla_{\mathbf{x}}$ is the del operator with respect to $\mathbf{x}$. Under mild condition, in the stationary state, $\frac{\partial p_l(\mathbf{x})}{\partial l} = 0$, so, the limit distribution $p(\mathbf{x}) = \lim_{l \to \infty} p_l(\mathbf{x})$ satisfies the below:

$$0 = -\nabla_{\mathbf{x}} \cdot \left(\frac{\tau}{2}\nabla_{\mathbf{x}}\left|\mathcal{R}_\theta(\mathbf{x})\right|^2 p(\mathbf{x})\right) + \beta^2\tau\left(\nabla_{\mathbf{x}} \cdot \nabla_{\mathbf{x}}p(\mathbf{x})\right) \tag{C.1.5}$$

This simplifies to steady state equation:

$$2\beta^2\nabla_{\mathbf{x}} \cdot \nabla_{\mathbf{x}}p(\mathbf{x}) = \nabla_{\mathbf{x}} \cdot \left(p(\mathbf{x})\nabla_{\mathbf{x}}\left|\mathcal{R}_\theta(\mathbf{x})\right|^2\right) \tag{C.1.6}$$

At first, we simplify the (RHS) of Equation (C.1.6). Since $\nabla_{\mathbf{x}}\left|\mathcal{R}_\theta(\mathbf{x})\right|^2 = 2\mathcal{R}_\theta(\mathbf{x})\nabla_{\mathbf{x}}\mathcal{R}_\theta(\mathbf{x})$, using the product rule, we can simplify as follows:

$$\nabla_{\mathbf{x}} \cdot \left(p(\mathbf{x})\nabla_{\mathbf{x}}\left|\mathcal{R}_\theta(\mathbf{x})\right|^2\right) = \nabla_{\mathbf{x}} \cdot \left(2p(\mathbf{x})\mathcal{R}_\theta(\mathbf{x})\nabla_{\mathbf{x}}\mathcal{R}_\theta(\mathbf{x})\right)$$
$$= 2\left(\mathcal{R}_\theta(\mathbf{x})\nabla_{\mathbf{x}}p(\mathbf{x}) \cdot \nabla_{\mathbf{x}}\mathcal{R}_\theta(\mathbf{x}) + p(\mathbf{x})\nabla_{\mathbf{x}}\mathcal{R}_\theta(\mathbf{x}) \cdot \nabla_{\mathbf{x}}\mathcal{R}_\theta(\mathbf{x}) + \mathcal{R}_\theta(\mathbf{x})p(\mathbf{x})\nabla_{\mathbf{x}} \cdot \nabla_{\mathbf{x}}\mathcal{R}_\theta(\mathbf{x})\right) \tag{C.1.7}$$

Substituting Equation (C.1.7) into the RHS of the Equation (C.1.6) and dividing by 2, we get:

$$\beta^2\nabla_{\mathbf{x}} \cdot \nabla_{\mathbf{x}}p(\mathbf{x}) = \mathcal{R}_\theta(\mathbf{x})\nabla_{\mathbf{x}}p(\mathbf{x}) \cdot \nabla_{\mathbf{x}}\mathcal{R}_\theta(\mathbf{x})$$
$$+ p(\mathbf{x})\nabla_{\mathbf{x}}\mathcal{R}_\theta(\mathbf{x}) \cdot \nabla_{\mathbf{x}}\mathcal{R}_\theta(\mathbf{x})$$
$$+ p(\mathbf{x})\mathcal{R}_\theta(\mathbf{x})\nabla_{\mathbf{x}} \cdot \nabla_{\mathbf{x}}\mathcal{R}_\theta(\mathbf{x}) \tag{C.1.8}$$

Now, if we assume the stationary distribution is of the form:

$$p(\mathbf{x}) = Z^{-1}\exp\left(c|\mathcal{R}_\theta(\mathbf{x})|^2\right) \tag{C.1.9}$$

where $Z$ is the partition function of the stationary distribution and $c > 0$ is the constant. Then, calculating the gradient and laplacian of Equation (C.1.9):

$$\nabla_{\mathbf{x}} p(\mathbf{x}) = 2cp(\mathbf{x})\mathcal{R}_\theta(\mathbf{x})\nabla_{\mathbf{x}}\mathcal{R}_\theta(\mathbf{x}), \tag{C.1.10}$$

$$\nabla_{\mathbf{x}} \cdot \nabla_{\mathbf{x}} p(\mathbf{x}) = \nabla_{\mathbf{x}} \cdot (2cp(\mathbf{x})\mathcal{R}_\theta(\mathbf{x})\nabla_{\mathbf{x}}\mathcal{R}_\theta(\mathbf{x})) \tag{C.1.11}$$

$$= 2c\mathcal{R}_\theta(\mathbf{x})\nabla_{\mathbf{x}} p(\mathbf{x}) \cdot \nabla_{\mathbf{x}}\mathcal{R}_\theta(\mathbf{x})$$
$$+ 2cp(\mathbf{x})\nabla_{\mathbf{x}}\mathcal{R}_\theta(\mathbf{x}) \cdot \nabla_{\mathbf{x}}\mathcal{R}_\theta(\mathbf{x})$$
$$+ 2cp(\mathbf{x})\mathcal{R}_\theta(\mathbf{x})\nabla_{\mathbf{x}} \cdot \nabla_{\mathbf{x}}\mathcal{R}_\theta(\mathbf{x}) \tag{C.1.12}$$

$$= 2c\mathcal{R}_\theta(\mathbf{x}) \left[ 2cp(\mathbf{x})\mathcal{R}_\theta(\mathbf{x})\nabla_{\mathbf{x}}\mathcal{R}_\theta(\mathbf{x}) \right] \cdot \nabla_{\mathbf{x}}\mathcal{R}_\theta(\mathbf{x})$$
$$+ 2cp(\mathbf{x})\nabla_{\mathbf{x}}\mathcal{R}_\theta(\mathbf{x}) \cdot \nabla_{\mathbf{x}}\mathcal{R}_\theta(\mathbf{x})$$
$$+ 2cp(\mathbf{x})\mathcal{R}_\theta(\mathbf{x})\nabla_{\mathbf{x}} \cdot \nabla_{\mathbf{x}}\mathcal{R}_\theta(\mathbf{x}) \tag{C.1.13}$$

$$= 4c^2 p(\mathbf{x})\mathcal{R}_\theta^2(\mathbf{x})\nabla_{\mathbf{x}}\mathcal{R}_\theta(\mathbf{x}) \cdot \nabla_{\mathbf{x}}\mathcal{R}_\theta(\mathbf{x})$$
$$+ 2cp(\mathbf{x})\nabla_{\mathbf{x}}\mathcal{R}_\theta(\mathbf{x}) \cdot \nabla_{\mathbf{x}}\mathcal{R}_\theta(\mathbf{x})$$
$$+ 2cp(\mathbf{x})\mathcal{R}_\theta(\mathbf{x})\nabla_{\mathbf{x}} \cdot \nabla_{\mathbf{x}}\mathcal{R}_\theta(\mathbf{x}) \tag{C.1.14}$$

Substituting these into the Equation (C.1.8), we get:

$$4c^2\beta^2 p(\mathbf{x})\mathcal{R}_\theta^2(\mathbf{x})\nabla_{\mathbf{x}}\mathcal{R}_\theta(\mathbf{x}) \cdot \nabla_{\mathbf{x}}\mathcal{R}_\theta(\mathbf{x})$$
$$+ 2c\beta^2 p(\mathbf{x})\nabla_{\mathbf{x}}\mathcal{R}_\theta(\mathbf{x}) \cdot \nabla_{\mathbf{x}}\mathcal{R}_\theta(\mathbf{x})$$
$$+ 2c\beta^2 p(\mathbf{x})\mathcal{R}_\theta(\mathbf{x})\nabla_{\mathbf{x}} \cdot \nabla_{\mathbf{x}}\mathcal{R}_\theta(\mathbf{x}) \tag{C.1.15}$$

$$= \mathcal{R}_\theta(\mathbf{x}) \left[ 2cp(\mathbf{x})\mathcal{R}_\theta(\mathbf{x})\nabla_{\mathbf{x}}\mathcal{R}_\theta(\mathbf{x}) \right] \cdot \nabla_{\mathbf{x}}\mathcal{R}_\theta(\mathbf{x})$$
$$+ p(\mathbf{x})\nabla_{\mathbf{x}}\mathcal{R}_\theta(\mathbf{x}) \cdot \nabla_{\mathbf{x}}\mathcal{R}_\theta(\mathbf{x}) + p(\mathbf{x})\mathcal{R}_\theta(\mathbf{x})\nabla_{\mathbf{x}} \cdot \nabla_{\mathbf{x}}\mathcal{R}_\theta(\mathbf{x}) \tag{C.1.16}$$

$$= 2cp(\mathbf{x})\mathcal{R}_\theta^2(\mathbf{x})\nabla_{\mathbf{x}}\mathcal{R}_\theta(\mathbf{x}) \cdot \nabla_{\mathbf{x}}\mathcal{R}_\theta(\mathbf{x})$$
$$+ p(\mathbf{x})\nabla_{\mathbf{x}}\mathcal{R}_\theta(\mathbf{x}) \cdot \nabla_{\mathbf{x}}\mathcal{R}_\theta(\mathbf{x}) + p(\mathbf{x})\mathcal{R}_\theta(\mathbf{x})\nabla_{\mathbf{x}} \cdot \nabla_{\mathbf{x}}\mathcal{R}_\theta(\mathbf{x}) \tag{C.1.17}$$

Since this equation holds for all $\mathbf{x}$ in the support of $p$, dividing through by $p(\mathbf{x})$ and simplifying:

$$\left(4c^2\beta^2 + 2c\beta^2\right)\mathcal{R}_\theta^2(\mathbf{x})\nabla_{\mathbf{x}}\mathcal{R}_\theta(\mathbf{x}) \cdot \nabla_{\mathbf{x}}\mathcal{R}_\theta(\mathbf{x}) + 2c\beta^2\mathcal{R}_\theta(\mathbf{x})\nabla_{\mathbf{x}} \cdot \nabla_{\mathbf{x}}\mathcal{R}_\theta(\mathbf{x})$$
$$= (2c+1)\mathcal{R}_\theta^2(\mathbf{x})\nabla_{\mathbf{x}}\mathcal{R}_\theta(\mathbf{x}) \cdot \nabla_{\mathbf{x}}\mathcal{R}_\theta(\mathbf{x}) + \mathcal{R}_\theta(\mathbf{x})\nabla_{\mathbf{x}} \cdot \nabla_{\mathbf{x}}\mathcal{R}_\theta(\mathbf{x}) \tag{C.1.18}$$

By comparing the coefficients of the terms on both sides, we get:

$$4c^2\beta^2 + 2c\beta^2 = 2c + 1, \tag{C.1.19}$$

$$2c\beta^2 = 1 \tag{C.1.20}$$

Thus, we can conclude $c = \frac{1}{2\beta^2}$, and the stationary distribution for the given modified Langevin dynamics satisfies:

$$p(\mathbf{x}) \propto \exp\left(\frac{|\mathcal{R}_\theta(\mathbf{x})|^2}{2\beta^2}\right) \tag{C.1.21}$$

# D SAMPLE TRAJECTORIES OF L-PINN AND BENCHMARK ALGORITHMS

We visualized the collocation point trajectories during the training under various adaptive sampling algorithms. The experimental settings follow the default configurations specified in the main text, with 4 layers and the number of Langevin iterations set to $l_{\mathrm{L}} = 1$. The background, shown as a heatmap using the plasma colormap, represents the residual landscape $|\mathcal{R}_\theta(\mathbf{x})|^2$, where dark purple indicates low values and bright yellow indicates high values. White points represent the collocation points used in training. Notably, the loss landscape dynamics remained largely consistent.

## D.1 RANDOM-R SAMPLE TRAJECTORY

The figure below represents the sample trajectory of Random-R, where different collocation points are uniformly sampled at each iteration.

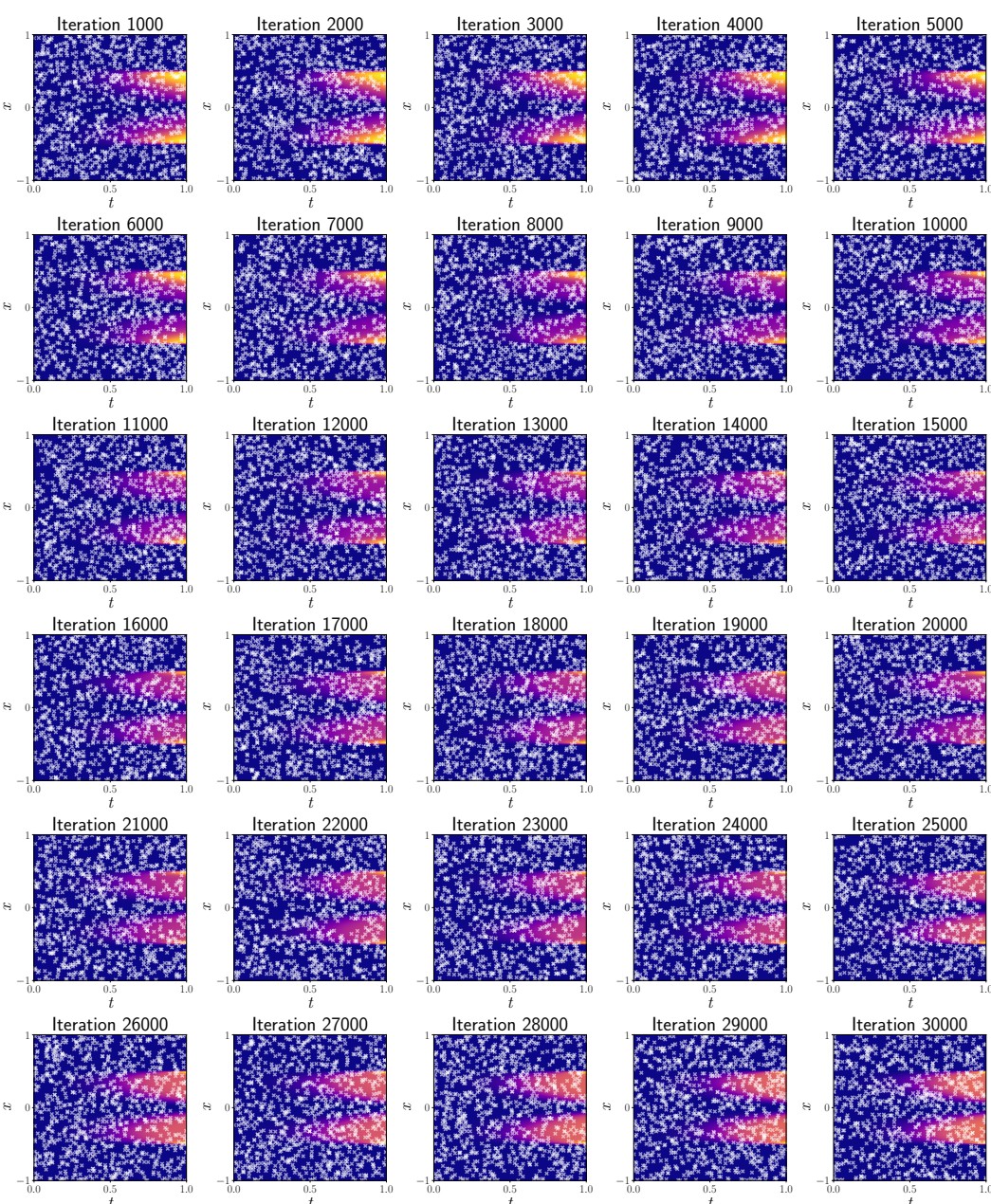

## D.2 RAD SAMPLE TRAJECTORY

This figure presents the RAD sample trajectory over multiple iterations, demonstrating a relatively stable pattern of sample distribution. As iterations progress, the sample points concentrate around regions of high residuals, with some diversity maintained throughout. However, despite the overall stability, the RAD sampling method exhibits a distribution that is not significantly different from the Random-R approach. The clustering becomes more pronounced in certain areas, but the overall spread and distribution of samples remain similar, suggesting that RAD does not offer a distinct advantage over random-R in terms of improving sampling diversity.

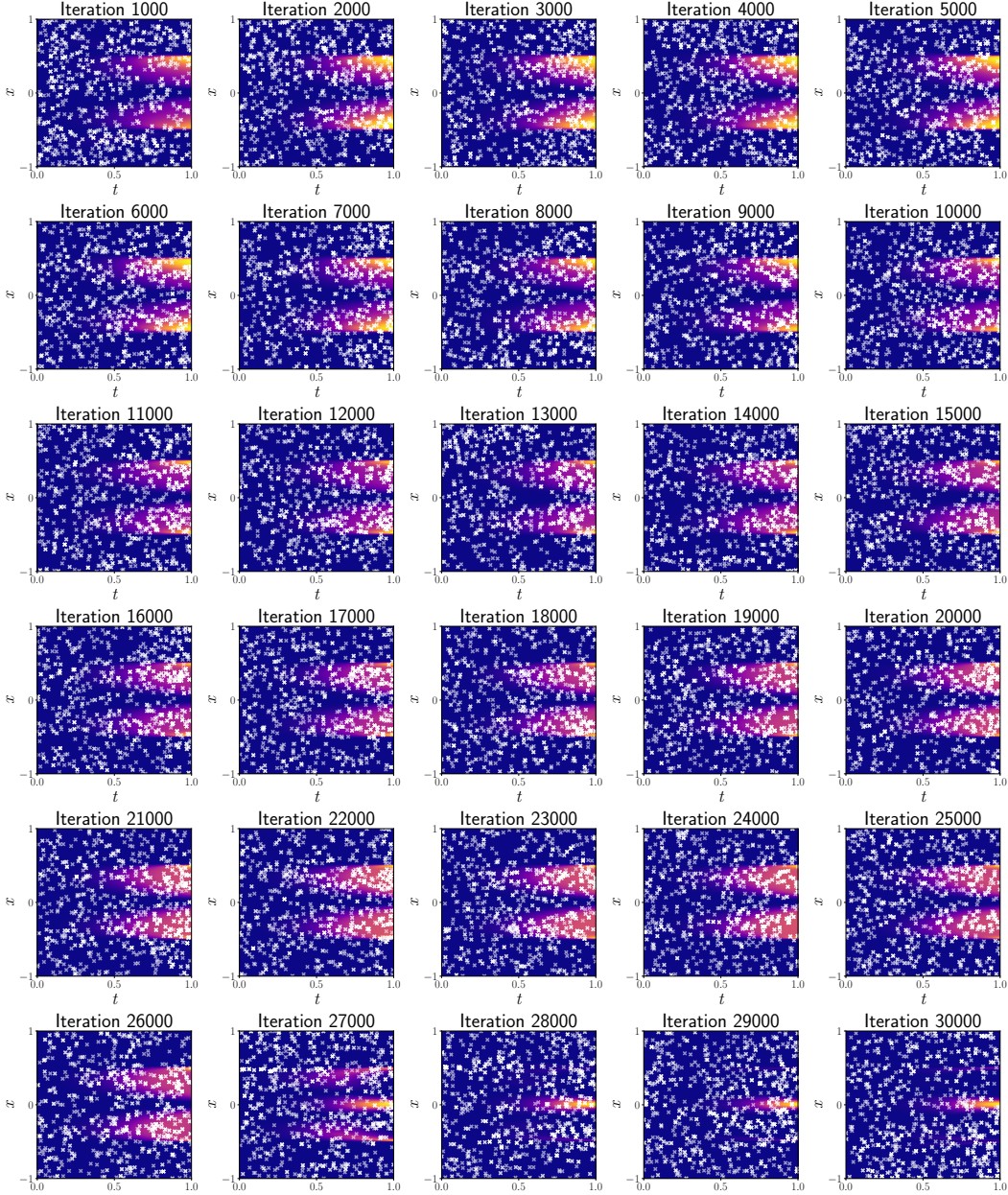

## D.3 R3 Sample Trajectory

This figure illustrates the evolution of sample trajectories in the R3 algorithm, showing a clear concentration of samples in regions with high residuals as the process progresses. While early iterations exhibit some scattering, the sample points increasingly cluster around specific areas of the residual landscape, leading to a lack of diversity in later stages. Furthermore, this imbalance indicates instability in the sampling strategy, as it fails to maintain a continuous, balanced shift in the sample population across the entire domain. The discontinuous change in the sample population may result in instability from the perspective of the learning process.

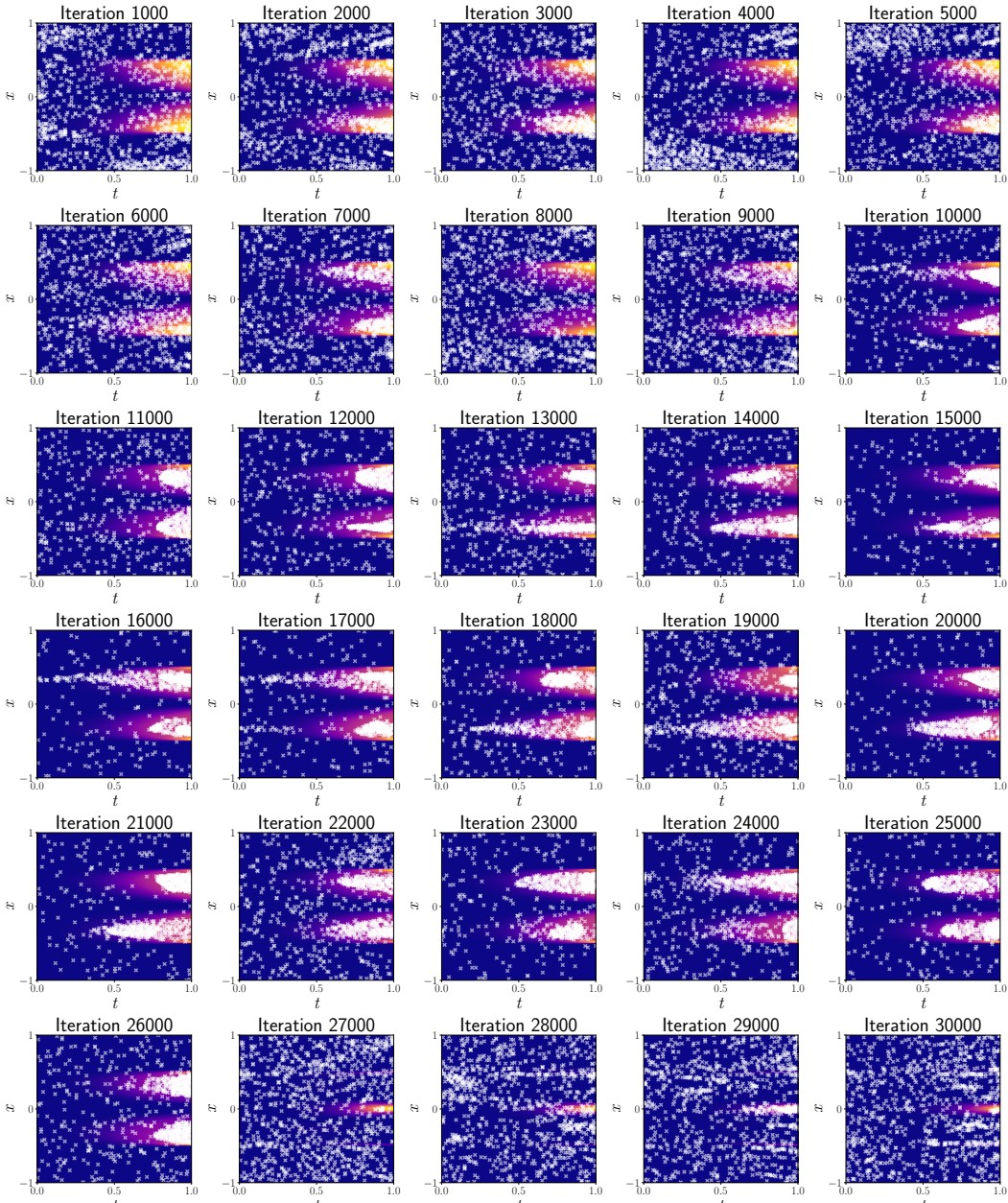

## D.4    $L^\infty$ Sample Trajectory

This figure illustrates the evolution of sample trajectories in the $L^\infty$ algorithm. As the number of iterations increases, the samples become overly concentrated in regions with high residuals, leading to a lack of diversity across the domain, particularly in areas with lower residuals. This imbalance goes against the goal of maintaining a well-distributed sample set proportional to the residual landscape. While some adaptation occurs, the excessive focus on extreme residuals (small $\beta$ case) results in a skewed distribution, highlighting the need for more balanced and diverse sampling to improve the algorithm's performance.

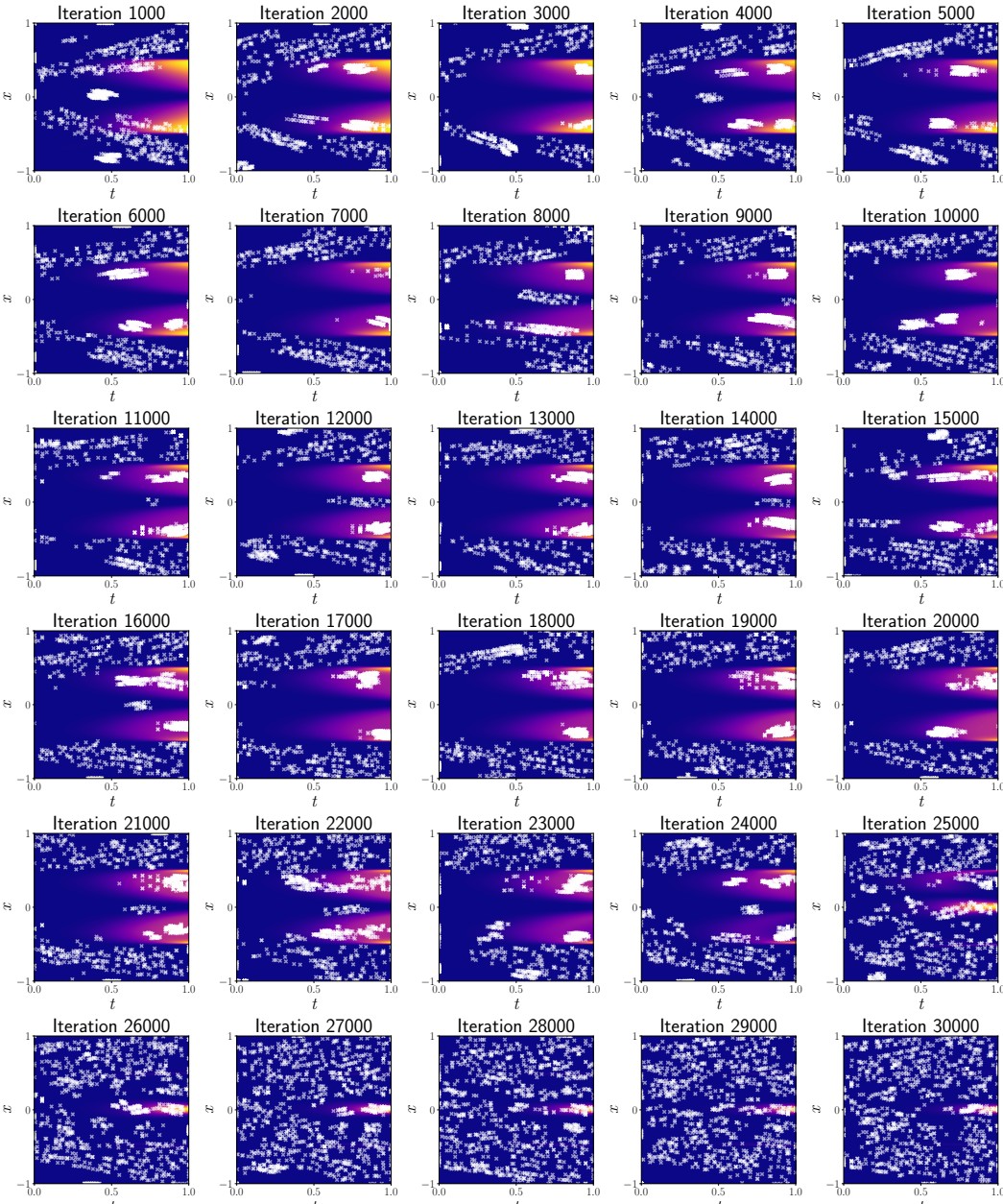

## D.5 L-PINN SAMPLE TRAJECTORY

This figure depicts the sample trajectory of the proposed L-PINN algorithm. As iterations progress, the sample points are proportionally distributed according to the residual landscape, maintaining diversity across the domain. Unlike other methods, proposed L-PINN algorithm avoids over-concentration in regions of high residuals, instead ensuring that sample points are scattered in a balanced manner. Additionally, the distribution adapts in line with the residual peaks, with an appropriate portion of samples allocated based on the peak heights. This indicates that the L-PINN algorithm successfully addresses the key objectives of both proportionality and diversity in sample distribution, improving stability and overall performance.

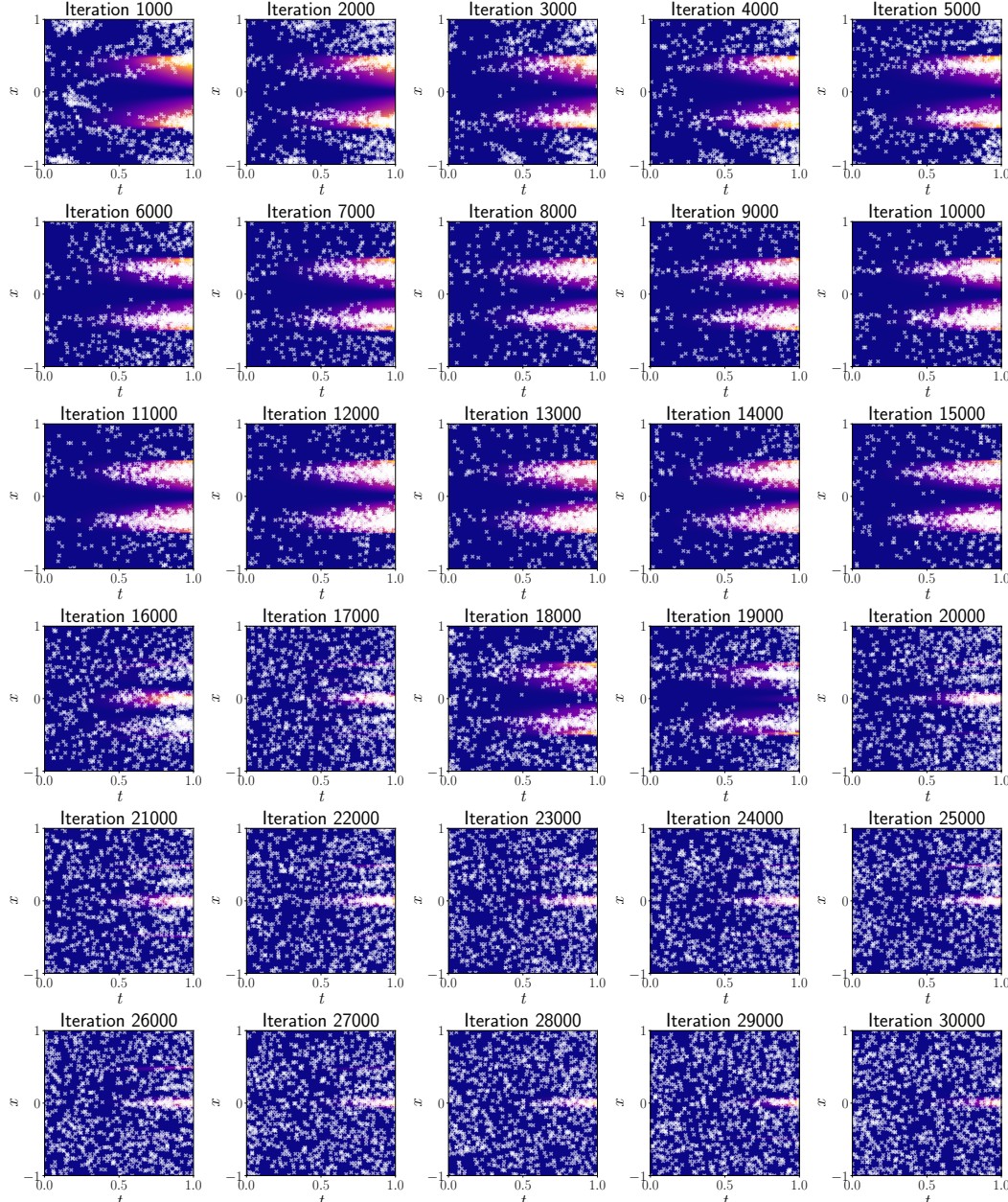

# E RELATIONSHIP BETWEEN LEARNING RATE AND MODEL COMPLEXITY

## E.1 LEARNING CURVE VARIATION WITH INCREASING DEPTH

Here, we aim to visualize and interpret the learning curves for the $\mathrm{Allen-Cahn}$ equation observed during the training process of the models, as reported in Table 1, with detailed experimental settings provided in Section 5, varying only the depth of the neural networks.

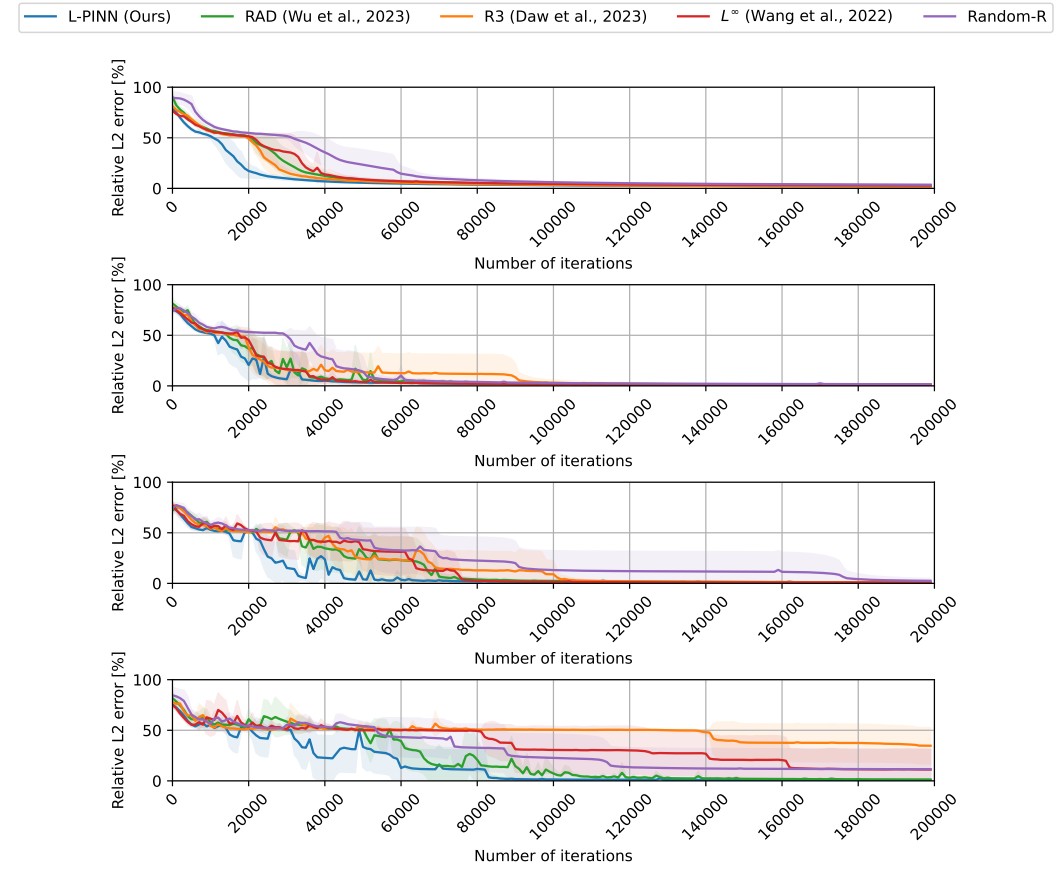

Figure 9: From top to bottom in the figure, the learning curves correspond to layers 4, 6, 8, and 10 for the $\mathrm{Allen-Cahn}$ equation.

Our primary observation is that most algorithms exhibit a slow learning progression until the learning rate reaches a specific value (which, of course, varies depending on the algorithm). This phenomenon appears to correlate with the degree of high residual concentration in the residual landscape of each algorithm. Specifically, the relative $L^2$ error in the learning curve requires more iterations to drop below a certain threshold (denoted as 50 in the figure) as the number of layers increases.

From an algorithmic perspective, most methods achieve the 50-threshold of the relative $L^2$ error crossing before iteration 40,000 with a 4 layer network. However, as the number of layers increases, particularly with 8 and 10 layers, the threshold-crossing iterations are significantly delayed. This delay is especially pronounced in algorithms such as R3 and $L^\infty$, which are highly focused on regions of extreme high residuals. This observation suggests that these algorithms are more affected by the increased complexity and residual concentration in deeper networks.

To verify whether this phenomenon depends on overall model complexity, in the following subsection, we also conducted experiments focusing on increasing the width rather than the depth of the model.

### E.2 Learning Curve Variation with Increasing Width

Simple calculations show that a neural network with 8 hidden layers and 128 nodes per layer has the same number of parameters as a neural network with 4 hidden layers and a width of 203. However, comparisons based solely on parameter count are inadequate, as depth introduces issues such as gradient vanishing.

Therefore, instead of viewing width solely from the perspective of parameter count, we conducted experiments by progressively doubling the width. The results showed that, similar to depth, increasing width also led to a gradual breakdown in learning stability. Consistent with the rankings observed in depth experiments, among adaptive sampling techniques, L-PINN reached the relative $L^2$ error threshold of 50 the fastest. Interestingly, Random-R demonstrated robustness in this setting, particularly with wide neural networks.

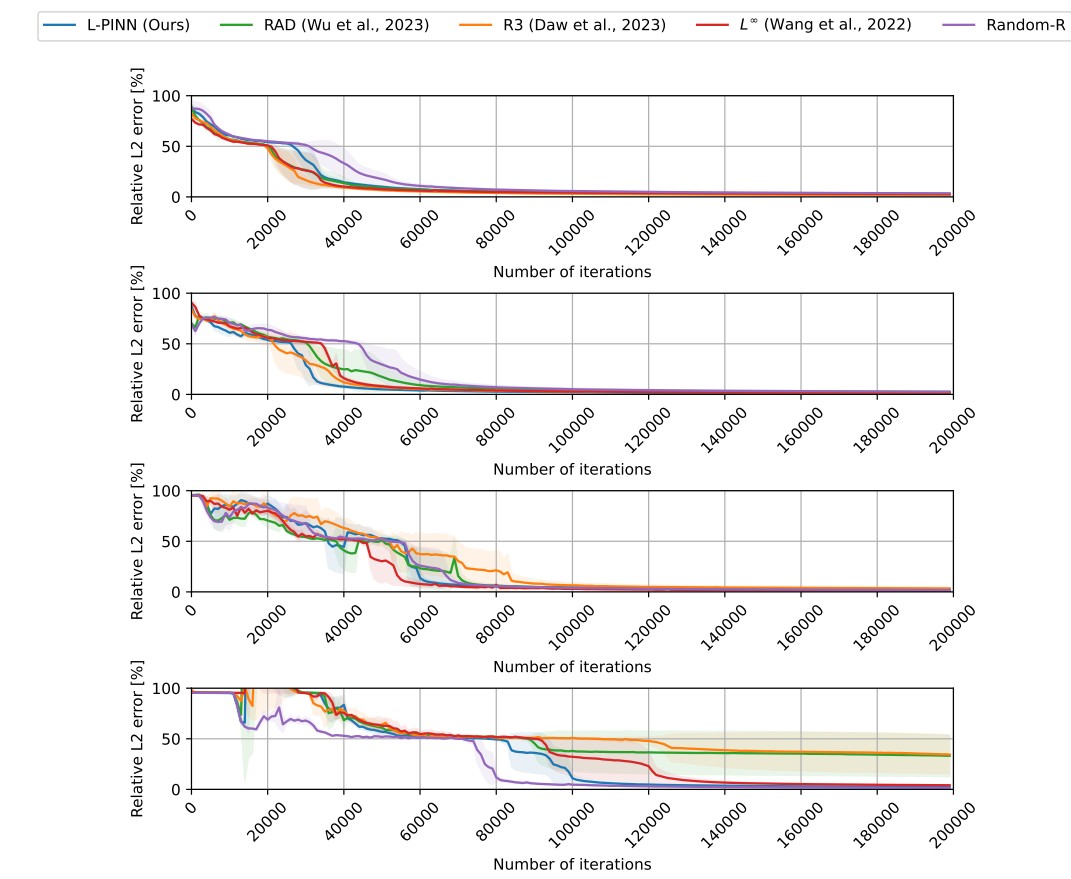

Figure 10: From top to bottom in the figure, the learning curves correspond to 128, 256, 512, and 1024 nodes per hidden layer, each with 4 layers, for the $\mathrm{Allen-Cahn}$ equation.

Through the experimental analyses described above, we argue that the proposed L-PINN demonstrates superior performance in terms of learning stability, particularly for models with high complexity.

# F SUPPLEMENTARY DETAILS ON EXPERIMENTAL SETUP

## F.1 DETAILS OF PARTIAL DIFFERENTIAL EQUATIONS

Burgers': We set $(\lambda_{\text{pde}}, \lambda_{\text{ic}}, \lambda_{\text{bc}}) = (1, 100, 1)$ to solve the equation

$$\frac{\partial u}{\partial t} + u\frac{\partial u}{\partial x} - \frac{0.01}{\pi}\frac{\partial^2 u}{\partial x^2} = 0, \quad x \in [-1, 1], \quad t \in [0, 1]; \tag{F.1.1}$$
$$u(-1, t) = u(1, t) = 0; \tag{F.1.2}$$
$$u(x, 0) = -\sin \pi x \tag{F.1.3}$$

Convection: We set $(\lambda_{\text{pde}}, \lambda_{\text{ic}}, \lambda_{\text{bc}}) = (1, 100, 100)$ to solve the equation

$$\frac{\partial u}{\partial t} + 50\frac{\partial u}{\partial x} = 0, \quad x \in [0, 2\pi], \quad t \in [0, 1]; \tag{F.1.4}$$
$$u(0, t) = u(2\pi, t); \tag{F.1.5}$$
$$u(x, 0) = \sin x \tag{F.1.6}$$

Allen−Cahn: We set $(\lambda_{\text{pde}}, \lambda_{\text{ic}}, \lambda_{\text{bc}}) = (1, 100, 1)$ to solve the equation

$$\frac{\partial u}{\partial t} - 0.0001\frac{\partial^2 u}{\partial x^2} - 5(u - u^3) = 0, \quad x \in [-1, 1], \quad t \in [0, 1]; \tag{F.1.7}$$
$$u(-1, t) = u(1, t); \tag{F.1.8}$$
$$u_x(-1, t) = u_x(1, t); \tag{F.1.9}$$
$$u(x, 0) = x^2 \cos \pi x \tag{F.1.10}$$

Korteweg−DeVries: We set $(\lambda_{\text{pde}}, \lambda_{\text{ic}}, \lambda_{\text{bc}}) = (1, 100, 1)$ to solve the equation

$$\frac{\partial u}{\partial t} + u\frac{\partial u}{\partial x} + 0.0025\frac{\partial^3 u}{\partial x^3} = 0, \quad x \in [-1, 1], \quad t \in [0, 1]; \tag{F.1.11}$$
$$u(-1, t) = u(1, t); \tag{F.1.12}$$
$$u(x, 0) = \cos \pi x \tag{F.1.13}$$

Schrödinger: We set $(\lambda_{\text{pde}}, \lambda_{\text{ic}}, \lambda_{\text{bc}}) = (1, 100, 1)$ to solve the equation

$$i\frac{\partial h}{\partial t} + 0.5\frac{\partial^2 h}{\partial x^2} + |h|^2 h = 0, \quad x \in [-5, 5], \quad t \in \left[0, \frac{\pi}{2}\right]; \tag{F.1.14}$$
$$h(-5, t) = h(5, t); \tag{F.1.15}$$
$$h_x(-5, t) = h_x(5, t); \tag{F.1.16}$$
$$h(x, 0) = 2\text{sech}(x) \tag{F.1.17}$$

## F.2 IMPLEMENTATION DETAILS OF BASELINE ALGORITHMS

For all PDEs, we conducted experiments by fixing the algorithms' hyperparameters to the values specified in the original baseline code. Specifically, for RAD, we set $c = k = 1$, and for $L^\infty$, we fixed the number of gradient steps at 20 and step size 0.05.

# G  ADDITIONAL RESULTS WITH VARYING L-PINN HYPERPARAMETERS

## G.1  VARIATION OF $\beta$ AND $l_L$ WITH FIXED $\tau = 0.002$

1. **Instability of performance for small $\beta$ values as the layer increases:** As the layer depth increases, small $\beta$ values lead to unstable performance. For instance, in layer 4, a small $\beta = 0.001$ results in a relatively stable error value of $1.18 \pm 0.23$ at $l_L = 1$, whereas in layer 10, the error rises significantly to $8.20 \pm 15.63$. This pattern suggests that small values of $\beta$ hinder performance stability in deeper layers.

2. **Increased instability with higher $l_L$ values:** Generally, the performance deteriorates as $l_L$ increases, particularly for small $\beta$ values. For example, in layer 6 with $\beta = 0.001$, the relative error increases from $0.58 \pm 0.08$ at $l_L = 1$ to $22.73 \pm 22.08$ at $l_L = 20$. This indicates that excessive Langevin iterations could lead to performance instability, especially when the concentration parameter $\beta$ is low.

| Layer and concentration parameter $\beta$ | | Langevin iteration $l_L$ | | | |
|---|---|---|---|---|---|
| Layer | $\beta$ | 1 | 5 | 10 | 20 |
| | 0.001 | $1.18 \pm 0.23$ | $1.63 \pm 0.40$ | $1.66 \pm 0.20$ | $6.34 \pm 2.69$ |
| | 0.05 | $1.53 \pm 0.36$ | $1.41 \pm 0.43$ | $1.27 \pm 0.21$ | $1.25 \pm 0.16$ |
| Layer 4 | 0.1 | $1.08 \pm 0.37$ | $\mathbf{0.93 \pm 0.09}$ | $1.23 \pm 0.23$ | $1.06 \pm 0.13$ |
| | 0.3 | $2.50 \pm 0.28$ | $2.30 \pm 0.34$ | $2.54 \pm 0.35$ | $2.51 \pm 0.36$ |
| | 0.4 | $2.80 \pm 0.60$ | $3.14 \pm 0.22$ | $3.14 \pm 0.49$ | $2.90 \pm 0.44$ |
| | 0.001 | $0.58 \pm 0.08$ | $10.89 \pm 19.65$ | $10.10 \pm 17.80$ | $22.73 \pm 22.08$ |
| | 0.05 | $0.62 \pm 0.11$ | $0.64 \pm 0.12$ | $0.65 \pm 0.12$ | $0.67 \pm 0.15$ |
| Layer 6 | 0.1 | $0.74 \pm 0.12$ | $0.58 \pm 0.08$ | $0.64 \pm 0.07$ | $\mathbf{0.58 \pm 0.03}$ |
| | 0.3 | $1.05 \pm 0.27$ | $1.15 \pm 0.21$ | $1.16 \pm 0.11$ | $1.33 \pm 0.13$ |
| | 0.4 | $1.47 \pm 0.10$ | $1.49 \pm 0.08$ | $1.48 \pm 0.22$ | $1.58 \pm 0.13$ |
| | 0.001 | $0.65 \pm 0.14$ | $1.25 \pm 0.73$ | $10.66 \pm 19.50$ | $22.14 \pm 24.51$ |
| | 0.05 | $0.88 \pm 0.34$ | $\mathbf{0.46 \pm 0.05}$ | $0.59 \pm 0.10$ | $1.44 \pm 0.81$ |
| Layer 8 | 0.1 | $0.59 \pm 0.14$ | $0.58 \pm 0.08$ | $0.68 \pm 0.35$ | $0.86 \pm 0.56$ |
| | 0.3 | $1.06 \pm 0.16$ | $1.15 \pm 0.21$ | $1.08 \pm 0.24$ | $0.92 \pm 0.11$ |
| | 0.4 | $1.12 \pm 0.25$ | $1.09 \pm 0.19$ | $1.14 \pm 0.17$ | $1.12 \pm 0.27$ |
| | 0.001 | $8.20 \pm 15.63$ | $23.14 \pm 20.74$ | $40.14 \pm 19.78$ | $41.79 \pm 20.50$ |
| | 0.05 | $\mathbf{0.46 \pm 0.17}$ | $17.33 \pm 21.03$ | $17.21 \pm 21.02$ | $30.08 \pm 24.07$ |
| Layer 10 | 0.1 | $0.63 \pm 0.20$ | $0.73 \pm 0.23$ | $0.70 \pm 0.22$ | $10.23 \pm 18.99$ |
| | 0.3 | $0.94 \pm 0.18$ | $0.98 \pm 0.15$ | $0.83 \pm 0.25$ | $1.02 \pm 0.30$ |
| | 0.4 | $0.94 \pm 0.15$ | $0.96 \pm 0.22$ | $0.97 \pm 0.37$ | $1.12 \pm 0.33$ |

Table 2: Relative $L^2$ error for varying $\beta$ and $l_L$ across different layers of the Allen−Cahn equation.

As a result, the main takeaway is that small values of $\beta$ are prone to instability as the layer depth increases and $l_L$ becomes large. Conversely, higher $\beta$ values can ensure stable performance, even with varying $l_L$ values. In shallow layers, however, lower $\beta$ values can be beneficial, providing a more precise error at lower $l_L$ values.

## G.2    Variation of $\tau$ and $l_{\text{L}}$ with Fixed $\beta = 0.2$

1. **Stability across $\tau$ values:** The relative $L^2$ error exhibits minor fluctuations across different values of $\tau$ for each layer suggesting a negligible dependency on $\tau$.

2. **Limited impact of $l_{\text{L}}$:** While increasing $l_{\text{L}}$ slightly reduces the variance of $L^2$ error in some cases, the effect is not consistent across layers showing only marginal improvement.

| Layer and Langevin step $\tau$ | | Langevin iteration $l_{\text{L}}$ | | | |
|---|---|---|---|---|---|
| Layer | $\tau$ | 1 | 5 | 10 | 20 |
| | 0.0001 | $1.98 \pm 0.35$ | $2.12 \pm 0.14$ | $\mathbf{1.66 \pm 0.06}$ | $1.74 \pm 0.19$ |
| | 0.0005 | $1.81 \pm 0.13$ | $1.74 \pm 0.40$ | $1.77 \pm 0.34$ | $1.74 \pm 0.35$ |
| Layer 4 | 0.001 | $2.35 \pm 0.37$ | $1.85 \pm 0.25$ | $1.98 \pm 0.18$ | $1.68 \pm 0.24$ |
| | 0.005 | $1.96 \pm 0.15$ | $2.08 \pm 0.18$ | $1.84 \pm 0.22$ | $1.71 \pm 0.17$ |
| | 0.01 | $1.93 \pm 0.28$ | $1.92 \pm 0.09$ | $1.48 \pm 0.09$ | $1.79 \pm 0.09$ |
| | 0.0001 | $0.96 \pm 0.11$ | $0.89 \pm 0.06$ | $0.96 \pm 0.11$ | $1.00 \pm 0.14$ |
| | 0.0005 | $0.92 \pm 0.13$ | $\mathbf{0.69 \pm 0.34}$ | $1.14 \pm 0.23$ | $1.17 \pm 0.25$ |
| Layer 6 | 0.001 | $0.85 \pm 0.07$ | $1.00 \pm 0.18$ | $0.70 \pm 0.09$ | $0.96 \pm 0.03$ |
| | 0.005 | $0.85 \pm 0.07$ | $0.82 \pm 0.22$ | $0.97 \pm 0.10$ | $0.78 \pm 0.37$ |
| | 0.01 | $0.90 \pm 0.04$ | $0.90 \pm 0.06$ | $1.03 \pm 0.16$ | $0.93 \pm 0.06$ |
| | 0.0001 | $\mathbf{0.62 \pm 0.04}$ | $0.79 \pm 0.11$ | $0.92 \pm 0.10$ | $0.82 \pm 0.05$ |
| | 0.0005 | $0.81 \pm 0.02$ | $0.85 \pm 0.12$ | $0.78 \pm 0.07$ | $0.75 \pm 0.06$ |
| Layer 8 | 0.001 | $0.91 \pm 0.13$ | $0.82 \pm 0.02$ | $0.71 \pm 0.04$ | $0.77 \pm 0.10$ |
| | 0.005 | $0.85 \pm 0.07$ | $0.83 \pm 0.04$ | $0.64 \pm 0.06$ | $0.63 \pm 0.08$ |
| | 0.01 | $0.64 \pm 0.14$ | $0.77 \pm 0.17$ | $0.88 \pm 0.13$ | $0.81 \pm 0.09$ |
| | 0.0001 | $\mathbf{0.41 \pm 0.18}$ | $0.75 \pm 0.08$ | $0.82 \pm 0.02$ | $0.68 \pm 0.12$ |
| | 0.0005 | $0.82 \pm 0.13$ | $0.64 \pm 0.04$ | $0.68 \pm 0.07$ | $0.82 \pm 0.06$ |
| Layer 10 | 0.001 | $0.91 \pm 0.21$ | $0.74 \pm 0.19$ | $0.87 \pm 0.25$ | $0.59 \pm 0.05$ |
| | 0.005 | $0.64 \pm 0.04$ | $0.53 \pm 0.32$ | $0.67 \pm 0.05$ | $0.64 \pm 0.12$ |
| | 0.01 | $0.56 \pm 0.29$ | $0.65 \pm 0.13$ | $0.71 \pm 0.14$ | $0.61 \pm 0.09$ |

Table 3: Relative $L^2$ error for varying $\tau$ and $l_{\text{L}}$ across different layers of the Allen$-$Cahn equation.

Overall, the impact of $\tau$ and $l_{\text{L}}$ on both relative $L^2$ error and variance is limited, indicating robustness of the method across a range of parameters. This robustness simplifies Langevin hyperparameter tuning, making the approach more practical for real-world applications.

## H  EXPERIMENTAL COMPARISON OF THE COMPUTATIONAL COMPLEXITIES

To evaluate computational complexities, we measured the computational costs for training deep neural networks using each algorithm. Specifically, to validate the scalability of the algorithms, we conducted experiments to analyze their computational requirements in terms of the number of collocation points $N_{\text{pde}}$ and the dimensionality of the PDE.

Based on our observations, the runtime of the sampling algorithms was independent of the specific PDE. Thus, we utilized equations that allowed for a straightforward extension from 1D to 2D in dimensionality. More specifically, we experimented with different sizes of collocation points (100, 1,000, 10,000, 50,000, and 100,000) for both 1D and 2D Burgers′ equations.

As part of the detailed experimental process, we calculated the elapsed time over 1,000 epochs. The measurement was repeated 10 times using 10 different random seeds, and the mean and standard deviation were computed. For additional clarity, the elapsed time was measured excluding auxiliary operations such as saving the model or storing data, focusing solely on the computations required to run the algorithms.

**Hardware specification.** NVIDIA RTX 4090 GPU with 24GB of memory.

**Changes with $N_{\textbf{PDE}}$.** As $N_{\text{PDE}}$ increases, the computational cost grows for all methods. However, the growth rate varies significantly between methods. Gradient-based algorithms such as L-PINN and $L^\infty$ show a particularly sharp increase in computational cost as $N_{\text{PDE}}$ grows. This is due to the iteration-intensive nature of their sampling processes. For example, with $N_{\text{PDE}} = 50,000$ in 2D, the computational cost of L-PINN ($l_{\text{L}} = 20$) reaches 613.96 seconds, whereas simpler methods like Fixed or Random-R remain below 35 seconds. For $N_{\text{PDE}} = 100,000$, L-PINN and $L^\infty$ run out of memory in the 2D case, highlighting their scalability limitations for very large PDE sample sizes.

**Changes with dimensionality (1D vs. 2D).** Extending from 1D to 2D consistently increases the computational cost for all methods. While simple methods like Fixed or Random-R exhibit a relatively modest increase in cost when transitioning from 1D to 2D, gradient-based methods such as L-PINN and $L^\infty$ show disproportionately higher computational times in the 2D case. For example, in the 2D case with $N_{\text{PDE}} = 1,000$, L-PINN ($l_{\text{L}} = 10$) takes 45.74 seconds, compared to only 16.97 seconds in 1D. At $N_{\text{PDE}} = 10,000$, L-PINN ($l_{\text{L}} = 10$) takes 59.32 seconds in 2D versus 20.43 seconds in 1D.

However, despite the overall computational expense of L-PINN for higher $l_{\text{L}}$ values, the case of $l_{\text{L}} = 1$ demonstrates significantly lower computational costs, making it relatively practical and scalable. For example, at $N_{\text{PDE}} = 50,000$, L-PINN ($l_{\text{L}} = 1$) takes 63.98 seconds in 2D, which is manageable compared to the prohibitive 324.49 seconds for $l_{\text{L}} = 10$. Similarly, for smaller $N_{\text{PDE}}$, such as 1,000, L-PINN ($l_{\text{L}} = 1$) shows competitive runtimes (e.g., 26.62 seconds in 2D). An additional advantage of using $l_{\text{L}} = 1$ is that it avoids the out of memory issues observed for higher values of $l_{\text{L}}$, even in large-scale scenarios such as $N_{\text{PDE}} = 100,000$ in 2D. Furthermore, the relative $L_2$ error reported in our paper uses $l_{\text{L}} = 1$ as the baseline, demonstrating its effectiveness in balancing computational efficiency with accuracy.

Finally, while gradient-based methods like L-PINN exhibit high computational costs due to the overhead of gradient computation, future work optimizing the gradient operations could significantly enhance the scalability and practicality of these methods. Thus, with improved optimization techniques, L-PINN ($l_{\text{L}} = 1$) remains a promising candidate for solving PDEs efficiently at scale.

Table 4: Comparison of computational complexities for Burgers' equation (1D and 2D), measured in seconds.

| Method | $N_{pde} = 100$ | | $N_{pde} = 1,000$ | | $N_{pde} = 10,000$ | |
|---|---|---|---|---|---|---|
| | 1D | 2D | 1D | 2D | 1D | 2D |
| Fixed | $2.58 \pm 0.03$ | $6.07 \pm 0.15$ | $2.54 \pm 0.03$ | $6.45 \pm 0.14$ | $3.07 \pm 0.05$ | $7.83 \pm 0.03$ |
| Random-R | $2.66 \pm 0.10$ | $6.22 \pm 0.21$ | $2.61 \pm 0.01$ | $6.54 \pm 0.16$ | $3.14 \pm 0.04$ | $7.85 \pm 0.03$ |
| R3 | $2.99 \pm 0.01$ | $6.15 \pm 0.19$ | $2.94 \pm 0.00$ | $6.57 \pm 0.22$ | $3.20 \pm 0.02$ | $7.97 \pm 0.03$ |
| RAD | $3.40 \pm 0.00$ | $7.61 \pm 0.02$ | $3.46 \pm 0.13$ | $8.00 \pm 0.09$ | $3.99 \pm 0.00$ | $10.04 \pm 0.01$ |
| L-PINN ($l_L = 1$) | $3.90 \pm 0.03$ | $9.96 \pm 0.26$ | $3.89 \pm 0.03$ | $10.42 \pm 0.31$ | $5.01 \pm 0.01$ | $13.59 \pm 0.03$ |
| L-PINN ($l_L = 5$) | $11.59 \pm 0.03$ | $25.67 \pm 0.51$ | $9.70 \pm 0.02$ | $26.62 \pm 0.37$ | $11.90 \pm 0.16$ | $33.40 \pm 0.14$ |
| L-PINN ($l_L = 10$) | $15.55 \pm 0.05$ | $44.68 \pm 0.52$ | $16.97 \pm 0.37$ | $45.74 \pm 0.92$ | $20.43 \pm 0.10$ | $59.32 \pm 0.27$ |
| L-PINN ($l_L = 20$) | $29.46 \pm 1.99$ | $85.10 \pm 0.15$ | $29.04 \pm 0.02$ | $85.40 \pm 1.19$ | $37.57 \pm 0.01$ | $110.21 \pm 0.17$ |
| $L^\infty$ | $28.77 \pm 1.99$ | $82.75 \pm 0.88$ | $29.03 \pm 2.20$ | $84.21 \pm 1.68$ | $37.54 \pm 0.48$ | $115.29 \pm 0.21$ |

| Method | $N_{pde} = 50,000$ | | $N_{pde} = 100,000$ | |
|---|---|---|---|---|
| | 1D | 2D | 1D | 2D |
| Fixed | $11.63 \pm 0.56$ | $34.83 \pm 0.04$ | $24.93 \pm 0.02$ | $80.17 \pm 0.05$ |
| Random-R | $11.66 \pm 0.05$ | $34.87 \pm 0.04$ | $24.91 \pm 0.04$ | $80.26 \pm 0.04$ |
| R3 | $11.97 \pm 0.04$ | $35.19 \pm 0.03$ | $25.34 \pm 0.05$ | $80.61 \pm 0.06$ |
| RAD | $14.92 \pm 0.03$ | $44.30 \pm 0.05$ | $31.81 \pm 0.09$ | $99.67 \pm 0.10$ |
| L-PINN ($l_L = 1$) | $21.19 \pm 0.06$ | $63.98 \pm 0.08$ | $46.36 \pm 0.05$ | $151.33 \pm 0.11$ |
| L-PINN ($l_L = 5$) | $56.86 \pm 0.06$ | $179.69 \pm 0.20$ | $130.95 \pm 0.08$ | Out of Memory |
| L-PINN ($l_L = 10$) | $102.52 \pm 0.17$ | $324.49 \pm 0.35$ | $236.95 \pm 0.11$ | Out of Memory |
| L-PINN ($l_L = 20$) | $192.52 \pm 0.30$ | $613.96 \pm 0.43$ | $448.62 \pm 0.30$ | Out of Memory |
| $L^\infty$ | $197.24 \pm 0.17$ | $612.22 \pm 0.53$ | $448.95 \pm 0.27$ | Out of Memory |

# I COMPATIBILITY OF L-PINN WITH HIGH-DIMENSIONAL PROBLEMS

To analyze the compatibility of L-PINN with high-dimensional problems, we first highlight its distinctions from existing adaptive sampling techniques, such as RAD. While both methods aim to achieve a balanced residual distribution for sampling, their approaches differ significantly. RAD utilizes Monte Carlo integration (MCI) to estimate $\mathbb{E}|\mathcal{R}_\theta(\mathbf{x})|^k$, whereas L-PINN bypasses this step by asymptotically converging to the desired distribution. However, this comes at the cost of increased hyperparameter complexity.

**Effect of high dimensionality on PDEs.** The difference between L-PINN and MCI-based methods becomes evident in high-dimensional PDEs with limited collocation points $N_{\text{pde}}$. For MCI, the accuracy depends on evenly distributed residual points, which becomes challenging as the spatial dimension increases. In contrast, L-PINN achieves the desired asymptotic distribution with fewer collocation points, provided that parameters $\tau$ and $l_\text{L}$ are properly tuned. For instance, in a 1D domain divided into $P$ uniform partitions, the probability of a sample falling into a specific partition is $1/P$. However, in higher dimensions, this value decreases exponentially, further complicating MCI-based approaches.

**Experimental results.** The limitations of adaptive sampling methods in high-dimensional settings are evident in the Heat equation experiments. Table 5 and Figure 11 summarize the results. For reproducibility, all experiments were conducted using default settings for both L-PINN and baseline methods. Detailed configurations are described in the main text section 5 and Appendix F.2. A notable observation is that Random-R performed better than other adaptive sampling techniques in the restricted 2D PDE cases we proposed. Nonetheless, it was observed that the proposed L-PINN consistently maintained the second-best performance, following Random-R, under default hyperparameter settings. Additionally, based on the visual results, it can be inferred that gradient-based algorithms such as $L^\infty$ and L-PINN captured high-frequency components at $t = 0$ more effectively than RAD or R3.

Table 5: Relative $L^2$ error comparison of methods for $\text{Burgers}'$ 2D and Heat 2D equations

| PDE | Method | | | | |
|---|---|---|---|---|---|
| | L-PINN | RAD | R3 | $L^\infty$ | Random-R |
| $\text{Burgers}'$ 2D | $0.05 \pm 0.00$ | $0.06 \pm 0.00$ | $0.06 \pm 0.00$ | $0.05 \pm 0.00$ | $0.05 \pm 0.00$ |
| Heat 2D | $1.03 \pm 0.26$ | $2.79 \pm 0.10$ | $9.14 \pm 1.18$ | $16.93 \pm 0.50$ | $0.43 \pm 0.03$ |

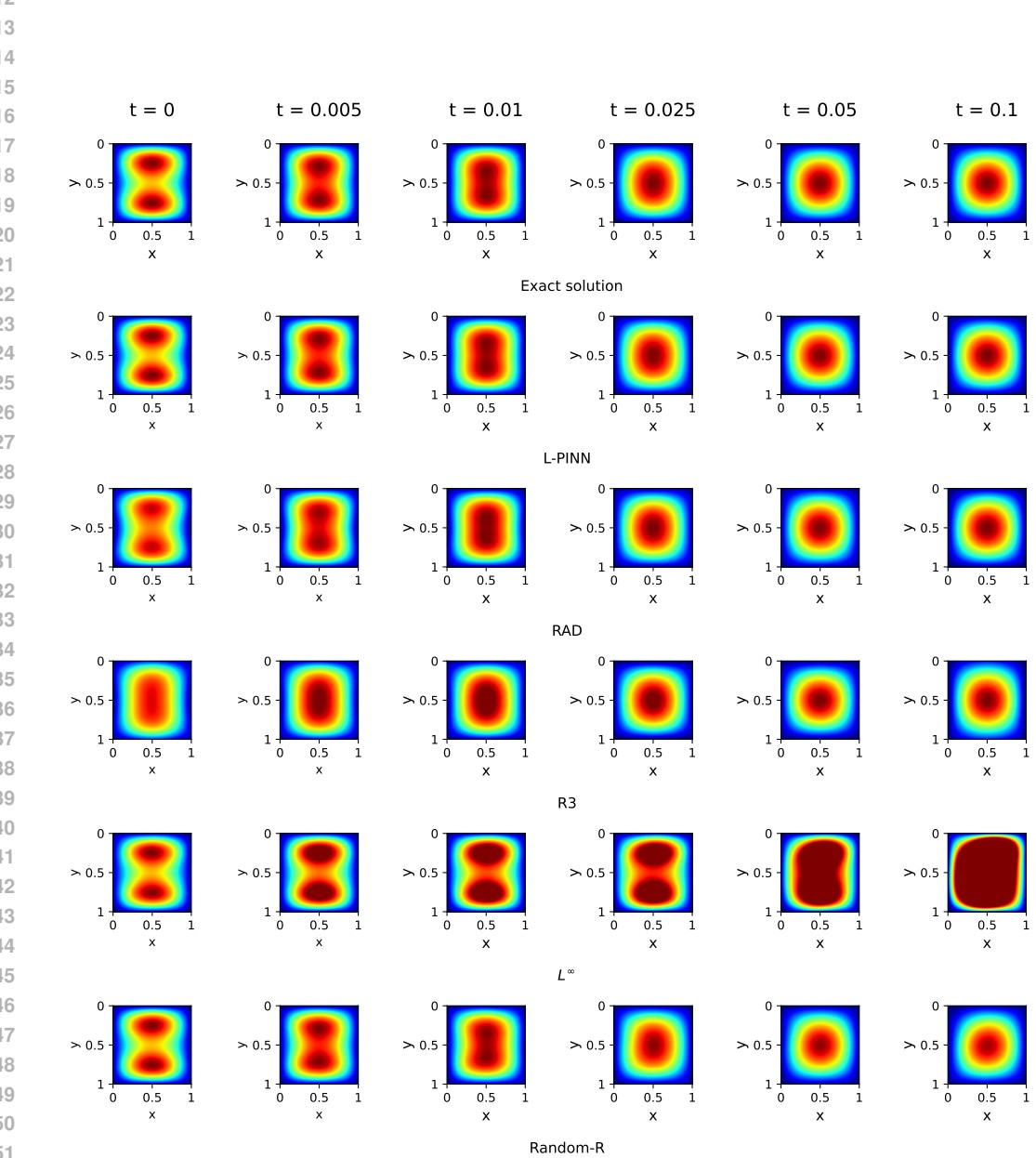

Figure 11: From top to bottom in the figure: exact solution, predicted solutions for benchmark algorithms, L-PINN, RAD, R3, $L^\infty$, and Random-R for the 2D Heat equation.

# J    COMPATIBILITY ISSUES WITH DIFFERENT NEURAL NETWORK ARCHITECTURES

We aimed to experimentally evaluate the compatibility of the proposed adaptive sampling technique with architectures beyond MLPs, including self-attention and modified-MLPs (Wang et al., 2023). Detailed descriptions of each architecture are provided in Table 6. Furthermore, to address the spectral bias issue commonly found in MLPs, we performed additional experiments incorporating random Fourier blocks (FB), as detailed in Tancik et al., 2020. The FB hyperparameters were set with a Fourier feature scale of 2 and a Fourier block dimension of 64.

Table 6: Parameter configuration for different architectures

| Parameter | MLP | Self-attention | Modified MLP |
|---|---|---|---|
| Activation | *Tanh* | *Tanh* | *Tanh* |
| Embedding dimension | 128 | 128 | 128 |
| Number of layers | 4 | 4 | 4 |
| Multi-head number | N/A | 4 | N/A |
| Fully connected dimension | N/A | 256 | N/A |
| Attention dropout | N/A | 0.1 | N/A |
| Additional encoders $U, V$ | N/A | N/A | *Yes* |

The results are summarized in Table 7. All algorithms were evaluated using the default settings provided in the benchmark algorithm papers, and the same applies to our approach, as detailed in the experimental settings described in the main text. Analyzing the results, we observe that the proposed L-PINN demonstrates high compatibility in terms of relative $L^2$ error. Even in the less favorable architectures, such as MLP and self-attention, L-PINN achieved the second-best performance. Notably, in scenarios incorporating FB, the proposed L-PINN consistently exhibited superior compatibility across all cases.

Table 7: Relative $L^2$ error comparison for different architectures across sampling methods.

| Architecture | L-PINN | RAD | R3 | $L^\infty$ | Random-R |
|---|---|---|---|---|---|
| MLP | $2.15 \pm 0.12$ | $2.65 \pm 0.47$ | $\mathbf{1.84 \pm 0.08}$ | $2.76 \pm 0.26$ | $3.56 \pm 0.20$ |
| MLP + FB | $\mathbf{0.56 \pm 0.14}$ | $0.69 \pm 0.05$ | $0.61 \pm 0.04$ | $0.81 \pm 0.18$ | $0.94 \pm 0.06$ |
| Modified MLP | $\mathbf{0.43 \pm 0.10}$ | $0.51 \pm 0.05$ | $0.66 \pm 0.07$ | $0.55 \pm 0.08$ | $0.56 \pm 0.07$ |
| Modified MLP + FB | $\mathbf{0.11 \pm 0.04}$ | $0.22 \pm 0.09$ | $0.24 \pm 0.03$ | $0.34 \pm 0.01$ | $0.21 \pm 0.04$ |
| Self-attention | $2.13 \pm 0.13$ | $2.30 \pm 0.13$ | $2.25 \pm 0.06$ | $\mathbf{1.98 \pm 0.03}$ | $2.91 \pm 0.69$ |
| Self-attention + FB | $\mathbf{1.29 \pm 0.09}$ | $1.53 \pm 0.03$ | $1.52 \pm 0.15$ | $1.33 \pm 0.12$ | $1.35 \pm 0.12$ |

## K  PSEUDO CODE LISTINGS

In this section, we provide the pseudo code for sampling via Langevin dynamics and the functions used in the training of the PINN model, specifically PDELoss(·) and Langevin_update(·). A notable feature is that the automatic partial derivative, originally used to compute the PDE loss, is applied once more for the Langevin update.

### K.1  PDE LOSS CALCULATION

```python
def PDELoss(DNN, XTGrid):
    u = DNN.forward(XTGrid)
    u_grad = torch.autograd.grad(
        outputs=u,
        inputs=XTGrid,
        grad_outputs=torch.ones(u.shape),
        create_graph=True,
        allow_unused=True
    )[0]

    ux, uy, ut = u_grad[:, 0], u_grad[:, 1], u_grad[:, 2]
    uxx = torch.autograd.grad(
        outputs=ux,
        inputs=XTGrid,
        grad_outputs=torch.ones(ux.shape),
        create_graph=True,
        allow_unused=True
    )[0][:, 0]

    uyy = torch.autograd.grad(
        outputs=uy,
        inputs=XTGrid,
        grad_outputs=torch.ones(uy.shape),
        create_graph=True,
        allow_unused=True
    )[0][:, 1]

    loss = (uxx + uyy - ut) ** 2
    return loss
```

Listing 1: PDE loss function

It can be observed that the PDE loss itself is not significantly different from the approach typically used in standard PINN models.

## K.2 LANGEVIN UPDATE

```
def Langevin_update(DNN, XYTGrid, l_\mathrm{L}=1, tau=2e-3, beta=0.2):
    for l in range(l_\mathrm{L}):
        loss = PDELoss(DNN, XYTGrid)
        XYT_grad = torch.autograd.grad(
            outputs=loss,
            inputs=XYTGrid,
            grad_outputs=torch.ones(loss.shape),
            create_graph=True,
            allow_unused=True
        )[0]

        scaler = torch.sqrt(torch.sum((XYT_grad + 1e-16) ** 2, axis=1)).
            reshape(-1, 1)
        XYT_grad = XYT_grad / scaler

        with torch.no_grad():
            XYTGrid += tau * XYT_grad + beta * torch.sqrt(2 * tau) *
                torch.randn(XYT_Grid.shape)
            XYTGrid[:, 0] = torch.clamp(XYTGrid[:, 0], min=0, max=1)
            XYTGrid[:, 1] = torch.clamp(XYTGrid[:, 1], min=0, max=1)
            XYTGrid[:, 2] = torch.clamp(XYTGrid[:, 2], min=0, max=0.1)
    return XYTGrid # updated grid points
```

Listing 2: Langevin update function

To explain the Langevin update function, additional details from the actual implementation are provided, with a key feature being the scaling of the Langevin gradient. As mentioned in the main text, this scaling is employed as a mechanism to ensure the stability of the training process. Furthermore, to prevent issues related to the feasibility of sample points due to the Langevin gradient update, we applied a clamp function to enforce the boundary condition. Finally, the collocation points are updated for $l_\mathrm{L}$ iterations using the step size $\tau$, the updated sample population is returned.

## K.3 TRAINING PROCESS OF 2D HEAT EQUATION

```
def Train(DNN, n_iters):
    x_init = torch.zeros(N_pde, 1, dtype=torch.float32).uniform_(0, 1)
    y_init = torch.zeros(N_pde, 1, dtype=torch.float32).uniform_(0, 1)
    t_init = torch.zeros(N_pde, 1, dtype=torch.float32).uniform_(0, 0.1)
    XYTGrid = torch.concatenate((x_init, y_init, t_init), axis=1)

    for i in range(n_iters):
        params = list(DNN.parameters())
        optimizer = torch.optim.Adam(params, lr=1e-3)
        XTYGrid = Langevin_update(DNN, XYTGrid)
        optimizer.zero_grad()

        pdeloss = PDELoss(DNN, XYTGrid)
        pdeloss.backward()
        optimizer.step()
```

Listing 3: Train function

The training process aligns exactly with the standard procedure for PINN models. First, the data is updated using Langevin dynamics, and then the updated data is used to compute the loss, followed by parameter updates.

