# OpenReview forum: "L-PINN: A Langevin Dynamics Approach with Balanced Sampling to Improve Learning Stability in Physics-Informed Neural Networks"
_ICLR.cc/2025/Conference — Submitted to ICLR 2025_

### Official Review · Reviewer_Zj8i · 2024-10-21

**Soundness:** 2
**Presentation:** 3
**Contribution:** 2
**Rating:** 5
**Confidence:** 3

**Summary:**

In this paper the authors highlight some potential shortcomings of residual-based methods for training PINNs and the lack of theoretical understanding thereof. In particular, the authors show theoretically that convergence requires a tighter upper bound on the learning rate. Furthermore, the authors propose a novel algorithm to train PINNs that exploits Langevin dynamics. The main idea is, instead of resampling the collocation points at each iteration (proportionally to the residual loss), to update existing sample position based on residual loss gradient.

**Strengths:**

- the paper is clearly written and easy to read.
- the idea of using Langevin dynamics to update sample points instead of resampling is elegant.

**Weaknesses:**

- while the theoretical contribution on the learning rate is interesting, it seems to be not very useful in practice. Based on the ablation study in Figure 3a and 4a it seems like that the steepness changes only slightly and that picking a smaller learning rate would already help convergence of current methods.
- furthermore, as the authors highlighted in the paper, Langevin dynamics comes at the price of two new additional parameters, which need to be fine tuned. As the authors themselves note, experiments in Appendix E suggest that the choice of these two parameters highly influence the performance of the proposed method.

**Questions:**

- Theorem 4.1 relies on an asymptotic limit for reaching collocation sample population. Therefore, I expected that performance would improve as the number of Langevin steps increases. Can you elaborate on why this is not the case based on the results in Appendix E? (the authors note that this behaviour is particularly true for small $\beta$ but also in the other cases it is hard to see an improvement if $l_L$ is increased). Somehow these results seem to imply that reaching collocation sample population is not useful in practice.
- It would be interesting to see quantitatively how much the choice of the step size $\tau$ in L-PINN affects convergence. It might be useful to have a comparison along the lines of the evaluation in Appendix E for $\beta$ and $l_L$, but in this case as $\tau$ varies.
- Instead of sampling new collocation points to find high-residuals, the proposed approach updates points according to the residual gradient. Intuitively, this seems to be more intense computationally, since sampling can usually be done rather cheaply. Did you run any experiments in this direction?
- I find Figure 5 a bit confusing. Could you clarify whether you are showing the error with respect to the solution or the solution itself?

Some minor typos:
- line 88: I would add $R_\theta(x)$ after "residual" such that it is clear what $R_\theta(x)$ is
- line 105: should use a comma instead of a point before "i.e."
- line 148: missing a point before "Additionally"
- line 430: for L-PINN, RAD, R3 and L* "exact solution" should be "predicted solution"
- line 460-466: results fro the Burgers' equation with 8 layers for Random-R and R3 should also be in boldface

---

> ### Author Response · Authors · 2024-11-20
>
> We appreciate your feedback on our paper. We have done our best to answer your keen questions.
>
> ---
>
> > **W1**
>
> We acknowledge that linking the importance of the learning rate to the flow of the text may create room for misinterpretation. However, the main claim of our paper regarding the learning rate is not "PINN training requires a higher or lower learning rate," but rather, "different algorithms prioritize concentration on high residuals differently, which, in turn, leads to stability issues depending on the learning rate and model complexity." We hope this point has been clarified.
>
> Additionally, if I interpret your concern correctly, it suggests, "Why not simply train with a lower learning rate?" This raises an intriguing question for us as well. To explore this further, we expanded the cases and visualized the results in **Fig. 4-(c)**. Based on our findings, a learning rate of at least 0.0005 appears necessary for various sampling methods to undergo a meaningful learning process.
>
> ---
>
> > **W2**
>
> You are correct that, in practice, the parameters that require tuning when implementing Langevin dynamics are $\tau$ and $l_L$. However, based on our experiments with the default Langevin hyperparameter settings ($l_L = 1$, $\tau = 0.002$) across various conditions, we observed robust performance for most PDE problems. This even held true for high-dimensional PDEs, as added in **Appendix I**. For the PDEs we examined, it does not seem necessary to have a large $l_L$, which is a highly desirable outcome from the perspective of computational complexity analysis.
>
> ---
>
> > **Q1**
>
> According to **Theorem 4**, the neural network $ f_\theta $ is fixed (and thus the residual landscape remains unchanged), with the conditions requiring $ l_L $ to be sufficiently large and **$ \tau$ to be sufficiently small**. From a practical implementation perspective, two factors must be considered:
> 1. The extent to which $f_\theta$, i.e., the residual landscape, varies over iterations. Empirically, based on sample trajectories, we observed relatively low temporal variation.
> 2. $l_L$ and $\tau$ must be inversely related, such that if $l_L$ increases, $\tau$ should decrease, and vice versa.
>
> Interpreting the results in **Appendix G**:
>
> The Langevin update equation $\mathbf{x}^{l+1}=\mathbf{x}^{l}+\frac{\tau}{2}\nabla_{\mathbf{x}}|\mathcal{R}_{\theta}(\mathbf{x}^{l})|^2+\beta\sqrt{\tau}z$ indicates that, under low $\beta$ (less exploration), performing multiple gradient updates with a relatively large step size $\tau$ compared to $l_L$ causes sample points to cluster around local modes, resulting in an overemphasis on high residuals. As demonstrated in our prior analysis, this leads to degraded stability. Furthermore, from the perspective of distributional convergence in Langevin dynamics, this imbalance between $l_L$ and $\tau$ also prevents convergence to the desired asymptotic population.
>
> Consequently, under relatively low $\beta$, if $l_L$ is relatively large compared to the step size $\tau$, sample points are drawn toward sharp modes in the residual landscape, inevitably resulting in a highly unstable learning process.
>
>
>
> ---
>
> > **Q2**
>
> As a follow-up to **Q1**, we are currently conducting experiments to compare the performance across varying $\tau$ and $l_L$ under fixed $\beta$ with proper balancing. However, due to resource constraints, we are unable to provide these results immediately. We will upload the findings as soon as they are completed.
>
> ---
>
> > **Q3**
>
> Experimental results related to computational complexity have been included in **Appendix H**.
>
> ---
>
> > **Q4**
>
> The visualization shows the absolute value of the gap between the exact solution and the predicted solution. There seems to have been a mistake in the description, leading to confusion. To avoid this, we have revised the figure caption for greater clarity.
>
> ---
>
> Lastly, we have incorporated all the minor comments you provided.

---

> > ### Author Response · Authors · 2024-11-25
> >
> > Dear Reviewer Zj8i:
> >
> > Regarding Q2, we have included additional experimental results in Appendix G.2.
> >
> > As the author-reviewer discussion period is nearing its conclusion, we kindly request you to review our responses at your earliest convenience. Should you have any additional questions or comments, we will make every effort to address them before the discussion period ends.
> >
> > We sincerely appreciate your time and valuable feedback. We look forward to hearing from you soon!
> >
> > Best regards,
> > ICLR 2025 Conference Submission1774 Authors

---

> > > ### Comment · Reviewer_Zj8i · 2024-11-25
> > >
> > > I would like to thank the authors for their answers and for running additional experiments, which should not be given for granted given the limited time for the rebuttal. And thanks for updating the pdf as well.
> > >
> > > As a general remark, I would suggest the authors (for future rebuttals) to consider color-coding the changes made to the pdf so that it is much clearer for reviewers where and what has been edited. I now compared the two pdfs and I have a clearer picture (so no need to do it now).
> > >
> > > Some further comments on the new results/edits:
> > > - I especially liked the new Figure 4, which now displays uncertainties as well. Looking at the old version, I think the plot was not very informative and possibly misleading, since the uncertainties reveal that some results the previously looked clearly separated are now compatible within the uncertainties (for instance results for learning rate 0.002 and 0.003)
> > >
> > > - With the new plots in Figure 4 I get the following insights: (i) a learning rate above 0.002 is too high for most methods, (ii) similarly, 0.0001 is too small and (iii) in the range 0.0001-0.001 the methods have comparable performance. I think my initial concern that a smaller learning rate would have been helpful for competing methods seems to be confirmed by the new results (previously it was explored only in the range 0.001-0.004). So I wouldn't say the proposed method results in better performance but rather that it is less sensitive to the (initial) learning rate (since anyways you are using Adam).
> > >
> > > - Why are the results for the relative L1 error for n_layers=10 in the new Figure1a different from the one you had before? I noticed that now you have a log-scale but this still doesn't explain the difference with the previous results. According to the new Figure 4a it is not true anymore that "As illustrated in Figure 4-(a), it can be observed that only L-PINN and RAD demonstrated stable performance when 10 hidden layers were used." Now every method except R3 is stable across architectures.
> > >
> > > - The computational complexity in Appendix H I think is very useful. My take-away is that the proposed approach is computationally viable as long as $l_L\leq5$, for which the runtime is already $\times 5$ that of Fixed/Random/R3/RAD.
> > >
> > > - Are the uncertainties in Figure 4 obtained by repeating the run with 5 different seeds as in Table 1? The same question applies to other results where $\pm$ results are reported (e.g. Table 5, 6). I would suggest to write this explicitly all the times repeated experiments are reported (if space allows, possibly directly in the caption).

---

> ### Author Response · Authors · 2024-11-25
>
> Dear Reviewer Zj8i,
>
> Thank you for your thoughtful and detailed feedback! Your insightful observations have been incredibly helpful in clarifying key aspects of our analysis.
>
> To address your main question, "Unlike the previous version, the results for $n_\text{layers} = 10$ appear more stable for $L^{\infty}$ and Random-R, similar to RAD," I would first like to note that we used 5 random seeds when generating **Figure 4**, just as we did for **Table 1**. In the earlier version, the representative values plotted in the figure were based on the **mean**. However, in the updated version, we used the **median** as the central metric for the boxplot. As a result, the values for $L^{\infty}$ and Random-R appear more stable than RAD.
>
> To provide a clearer understanding, we have included the raw results for $n_\text{layers} = 10$ based on the 5 seeds:
>
> - L-PINN: [1.03728259, 1.39095485, 1.17176408, 0.93006007, 0.76291417], mean = 1.058, median = 1.037
> - R3: [1.58583503, 49.7361213, 35.45759022, 35.65890789, 49.9127984], mean = 34.470, median = 35.658
> - RAD: [1.65305007, 1.31262923, 1.17702512, 1.15792109, 1.5187026], mean = 1.363, median = 1.312
> - **$L^{\infty}$**: [49.25066531, 1.20032271, 2.64070798, 0.89049591, 0.77396245], **mean = 10.951, median = 1.200**
> - **Random-R**: [1.08244075, 1.47837037, 51.34871602, 2.19732579, 1.34252282], **mean = 11.489, median = 1.478**
>
> As shown above, the discrepancy between the mean and the median caused some confusion, particularly for $L^{\infty}$ and Random-R. Additionally, to enhance the clarity of the plots, we generated the boxplots containing outlier points.
>
> Your observation was spot-on and helped us identify and address this issue. To reduce potential misunderstandings for readers, we have updated the figure caption to explicitly mention the use of repeated seeds and added a note about random seeds at the beginning of the ablation study section. Furthermore, we clarified this aspect by incorporating color coding (blue) in the updated version.
>
> Additionally, we strongly resonate with your take-home messages (regarding learning rate and computational feasibility), and we are glad that our intentions have been effectively communicated.
>
> Once again, thank you for your kind and perceptive comments—they have greatly enhanced the clarity of our work. If you have any further questions or concerns, please do not hesitate to leave a comment!
>
> Best regards,
> ICLR 2025 Conference Submission1774 Authors

---

> ### Author Response · Authors · 2024-11-27
>
> Dear Reviewer Zj8i:
>
> With less than 24 hours remaining in the rebuttal period, we kindly request your feedback on our responses. Additionally, we would greatly appreciate it if you could briefly indicate whether our replies sufficiently addressed your concerns to the extent that it might influence your decision positively, or if there will be no change in your evaluation. Thank you in advance for your time and consideration!
>
> Best regards,
> ICLR 2025 Conference Submission1774 Authors

---

> > ### Comment · Reviewer_Zj8i · 2024-11-27
> >
> > I thank the authors for updating the box plot, which I now think is much clearer.
> >
> > Do you know what is the origin of the outliers? It could be that the learning rate used is too big and the loss spiked towards the end of training (e.g. for $L^\infty$ and Random-R) or that it was too small and never improved (e.g. R3). Anyways, I believe that it is not something related to the specific methods themselves but just about their training.
> >
> > As mentioned in the previous response, I believe that now it is no longer true that "As illustrated in Figure 4-(a), it can be observed that only L-PINN and RAD demonstrated stable performance when 10 hidden layers were used.". All competing methods and the proposed approach show stable performances across different number of layers.
> >
> > Overall, I believe that the proposed approach provides an interesting alternative to re-sampling methods with Langevin dynamics. Experimentally, this shows to provide less sensitivity to the learning rate. However, I do not see other advantages of the proposed method compared to other methods. For this reasons I would like to keep my score as is.

---

> ### Author Response · Authors · 2024-11-27
>
> Dear Reviewer Zj8i,
>
> Thank you for your insightful comments and observations. Below, we provide a detailed response to your points:
>
> First, we select the model's performance based on the point in the training process where the overall loss curve reaches its lowest value. In other words, the reported performance is not based on specific values from sudden spikes in the loss. This behavior is clearly illustrated in the learning curve presented in **Appendix E.2**, specifically in the bottom of **Figure 10**.
>
> > **Question:** Do you know what is the origin of the outliers? It could be that the learning rate used is too big and the loss spiked towards the end of training (e.g. for $L^{\infty}$ and Random-R) or that it was too small and never improved (e.g. R3). Anyways, I believe that it is not something related to the specific methods themselves but just about their training.
> ---
>
> Regarding the experimental setup, all algorithms utilized a learning rate of $\eta = 0.001$, with a scheduler multiplying the learning rate by 0.9 every 5000 iterations. Our observations indicate that for the Allen-Cahn equation, the ability to drop below a threshold of 50 serves as a reliable benchmark for stable convergence. However, as layer depth increases, the number of iterations required to meet this threshold also grows, as detailed in **Appendix E.2, Figure 10**. This trend is particularly pronounced for $L^{\infty}$ and R3, while L-PINN and RAD demonstrate relatively greater robustness under these conditions. Ultimately, this implies that for deeper networks, the learning rate must decay sufficiently before the relative $ L^2 $ error can drop below 50, which takes a significant number of iterations and can be interpreted as the cause of outliers. The reason why this becomes a cause of outliers will become clearer as we address the part you commented on below.
>
> > **Comment:** As mentioned in the previous response, I believe that now it is no longer true that "As illustrated in Figure 4-(a), it can be observed that only L-PINN and RAD demonstrated stable performance when 10 hidden layers were used.". All competing methods and the proposed approach show stable performances across different number of layers.
>
> We also agree with your hypothesis that "a smaller learning rate might enable unstable algorithms to learn effectively even with deep layers." This aligns with our interpretation of your comment, suggesting that "if learning rates are appropriately adjusted as layers deepen, stability could be ensured for all algorithms." However, we note a potential issue with this approach: while a lower learning rate may enable earlier convergence below the threshold of 50, **it could also result in diminished learning ability during the remaining iterations due to reduced learning rates.** Although regarding this discussion pertains to the case where the number of layers is 4 without scheduler, the limitation arising from naively lowering the learning rate is evident in the results presented in **Figure 4-(c)** of our paper.
>
>
> > **Concern:** However, I do not see other advantages of the proposed method compared to other methods.
> ---
>
> Based on the responses to the question and comment, we believe that the stability of our proposed L-PINN lies in its reduced sensitivity to both layer depth and learning rate, which we consider a significant strength of our algorithm. That said, we fully acknowledge the importance of the issue you raised and are committed to exploring it further.
>
> ---
>
> To address your concerns, we are planning to conduct additional experiments by setting the layer depth to 10 and using an initial learning rate of 0.0005. Furthermore, to provide better clarification regarding whether the occurrence of outliers is due to randomness, we will expand the experiments shown in **Figure 4-(a)** by adding five more seeds for the case of layer 10. We hope these efforts will help resolve any remaining uncertainties. However, due to the time required for additional experiments, we will upload these results as soon as they are completed, aiming to do so before December 2nd or 3rd, within the discussion period specified by ICLR, to enable further discussion based on your and our comments.
>
> If you have additional questions or comments about these experiments, please do not hesitate to reach out. Once again, thank you for your valuable insights.
>
> Best regards,
> ICLR 2025 Conference Submission1774 Authors

---

> > ### Comment · Reviewer_Zj8i · 2024-11-27
> >
> > I thank the authors for the follow-up response.
> >
> > Concerning the first question, I think now the discussion is becoming quite focused on the precise learning rate and scheduler used. I think that it might have an influence but it is not the most interesting aspect here, at least in my opinion.
> >
> > My insight from Figure 4a is that given a fixed learning rate (and scheduler) all methods perform similarly well across number of layers (except for R3 with 10 layers). The result you had before showed worse performance on the mean value (instead of the median) because of one outlier. This was the reason that originally motivated the sentence "As illustrated in Figure 4-(a), it can be observed that only L-PINN and RAD demonstrated stable performance when 10 hidden layers were used", which however I think now it is not supported by the updated results.
> >
> > Maybe your insight was coincidentally correct and your algorithm is indeed more stable across different number of layers, but at the moment I don't think there is evidence supporting the claim.
> >
> > I don't believe that additional experiments with a different learning rate will be particularly enlightening for this matter, but of course I might be wrong.

---

> ### Author Response · Authors · 2024-11-27
>
> Dear Reviewer Zj8i,
>
> Thank you for the prompt feedback. Your summary has greatly clarified the direction we need to focus on for further discussion.
>
> First, I believe the notion of **"stable performance"** must be clearly defined. Based on my understanding of your argument:
>
> In the earlier version (without raw results based on 5 seeds), when evaluating the results based on the **mean**, it seemed that the error across all seeds would be approximately 10–11. In this case, the statement, *"As illustrated in Figure 4-(a), it can be observed that only L-PINN and RAD demonstrated stable performance when 10 hidden layers were used,"* was valid. However, with the updated results based on the **median**, when excluding outliers, this statement seems invalid because performance is similar across algorithms.
>
> I think this is the point where we need to reach a consensus. To summarize, the experimental results so far, we can categorize them into three cases:
> 1. exhibits failures and does not achieve desirable performance (R3),
> 2. occasionally exhibit failures but achieve desirable performance ($L^{\infty}$ and Random-R),
> 3. exhibit no failures and achieve desirable performance (L-PINN and RAD).
>
> It seems that your interpretation might be grouping categories 2 and 3 together to denote **stable performance**, whereas, in our case, the term **stable performance** refers to the third category only.
>
> If our understanding is correct, would it be helpful to expand the experiments by increasing the number of seeds for the 10-layer case as evidence supporting the claim? This might allow us to more accurately measure the frequency of failures for each algorithm.
>
> Once again, thank you for your swift feedback and valuable comments. If you have further questions or suggestions, please do not hesitate to share them!
>
> Best regards,
> ICLR 2025 Conference Submission1774 Authors

---

> > ### Comment · Reviewer_Zj8i · 2024-11-29
> >
> > Honestly, I think that the statement "As illustrated in Figure 4-(a), it can be observed that only L-PINN and RAD demonstrated stable performance when 10 hidden layers were used," was a bit stretched also before, since it was anyway tested on a single dataset and the trend was shown only for 10 layers while for other number of layers all methods are comparable. On the other hand, results over different learning rates show stable performance over a wider range of values.
> >
> > Therefore, I don't believe that experiments with 10 seeds would dramatically change our insights. My current understanding (also from this experiment with different number of layers) is that the proposed approach is more stable across different choices of initial learning rates.

---

> > > ### Author Response · Authors · 2024-11-30
> > >
> > > Dear Reviewer Zj8i,
> > >
> > > Thank you for your insightful comments regarding the statement: *"As illustrated in Figure 4-(a), it can be observed that only L-PINN and RAD demonstrated stable performance when 10 hidden layers were used."* We agree with your observation that for smaller architectures (e.g., 4 and 6 hidden layers), the methods exhibit comparable stability. As for the claim that instability increases with the number of layers, we acknowledge that this observation alone may seem a bit stretched.
> > >
> > > However, regarding the accuracy of the statement, our original intent was to highlight the phenomenon observed specifically under the condition of using 10 hidden layers, where most algorithms exhibited instability, while only L-PINN and RAD demonstrated what we defined as "stable performance" (category 3, as defined above). If the statement had been generalized to claim instability "as the number of layers increases" or lacked the explicit reference to "10 hidden layers," we agree that it could have been interpreted as overstated. However, since the statement explicitly references the observed behavior at 10 hidden layers, we believe it remains valid within its specific context.
> > >
> > > Additionally, we appreciate your observation regarding learning rate trends, which further highlight L-PINN's consistent stability across a wide range of values. This observation reinforces our key contribution in analyzing the interplay of model complexity and learning rates on training stability. The robustness of L-PINN under varying learning rates underscores its practical utility and aligns closely with the broader objectives of this study.
> > >
> > > While we are unable to revise the manuscript at this stage, we will ensure this clarification is incorporated in the final version of the paper should it be accepted. We hope this response provides sufficient clarity regarding the intended scope of our claims.
> > >
> > > Best regards,
> > > ICLR 2025 Conference Submission1774 Authors

---

> > > > ### Comment · Reviewer_Zj8i · 2024-12-01
> > > >
> > > > Thanks for continuing the discussion. I agree that the statement is factually true (for these 5 seeds) but my impression while reading the paragraph was somehow that the proposed method is more robust to an increase in architecture complexity, which however is limited to 10 layers (and now less evident when considering the median instead of the mean). Maybe this impression was given by the next sentence "These results indicate that high residual methods are more susceptible to increasing model complexity, whereas L-PINN remains robust." which I think it's a bit stretched.
> > > >
> > > > I would recommend the authors to adapt these two sentences such that the claim is clearly restricted to the specific case under study. And I would agree that using more seeds would make the intended statement clearer (but still won't make it more general).

---

> > > > > ### Author Response · Authors · 2024-12-02
> > > > >
> > > > > Dear Reviewer Zj8i,
> > > > >
> > > > > Although we may not have directly addressed the issue in question, we are glad that we were able to partially address the points of discomfort you raised. Thank you for your continued interest and thoughtful attention! We will make additional adjustments based on the final comments you provided.
> > > > >
> > > > > Best regards,
> > > > > ICLR 2025 Conference Submission1774 Authors

---

### Official Review · Reviewer_SkJ6 · 2024-10-26

**Soundness:** 3
**Presentation:** 3
**Contribution:** 3
**Rating:** 6
**Confidence:** 4

**Summary:**

This paper examines the relationship between sampling strategies and learning stability in PINNs. The authors provide theoretical analysis showing that sampling methods focused on high-residual points require stricter learning rate constraints for stability, especially with increased model complexity. They present a Langevin dynamics-based PINN (L-PINN) framework that implements balanced sampling proportional to PDE residuals. They vaildate the effectiveness of the proposed sampling method across multiple PDEs, showing that  L-PINN achieves comparable or better relative L2 error performance while maintaining stability across different model complexities and learning rates compared to existing methods.

**Strengths:**

1.  The proposed method is novel and provides rigorous theoretical analysis of stability issues in high-residual sampling methods. The authors establish clear mathematical connections between sampling strategies and learning rate constraints, backing their theoretical claims with detailed proofs. This helps explain the stability challenges that emerge as model complexity increases.

2. Another strength is the paper's thorough empirical validation. The experiments span multiple representative PDEs (Burgers', Convection, Allen-Cahn). The authors carefully compare their approach against various sampling methods, with comprehensive ablation studies on learning rates and network depth.

**Weaknesses:**

1.  The assumption 3.1 seems to be too strong for nonlinear PDEs. But I think it is ok for linear PDEs.

2. Section 5.2's experimental results are dense and difficult to follow - this content could be better organized with supporting details moved to the appendix.

3.  The authors overlook relevant prior work, particularly PirateNet [1], which addresses similar scaling challenges in training deep PINN models through adaptive skip connections.

[1] Wang, S., Li, B., Chen, Y. and Perdikaris, P., 2024. PirateNets: Physics-informed Deep Learning with Residual Adaptive Networks. arXiv preprint arXiv:2402.00326.

**Questions:**

1. What exactly are the feature vectors being visualized in Figure 6? The paper relies on Assumption 3.1 to express PDE residuals as linear combinations of feature-mapped vectors, but does not clearly define how these feature vectors ϕ(x) are obtained for nonlinear PDEs. Are these simply outputs from hidden layers, or do they incorporate PDE residual information? The authors should provide precise mathematical expressions for the visualized quantities.

2.  If these feature vectors are merely hidden layer outputs without considering PDE residuals, there appears to be a significant gap in the paper's logic. How do these empirical visualizations justify Assumption 3.2 about heavy-tailed distributions of feature vectors that appear in Eq 3.1?  This disconnect between the theoretical framework and empirical validation needs to be addressed.

---

> ### Author Response · Authors · 2024-11-20
>
> Thank you for your valuable review and constructive suggestions. We have addressed your comments point by point below.
>
> ---
>
> > **W1**
>
> I also generally agree with the validity of the assumptions from an analytic perspective. However, I believe that these issues are somewhat mitigated during the learning process, leveraging the representation power of neural networks. A more detailed discussion on this topic is provided in **Q1**.
>
> ---
>
> > **W2**
>
> The PDE experimental settings for Section 5.2 have been moved to **Appendix F**.
>
> ---
>
> > **W3**
>
> This was a paper we had not previously reviewed, but upon examination, we found it to be a meaningful work with motivations closely aligned with ours. As such, it has been added as a reference addressing scalability with respect to model complexity.
>
> That said, while this paper approaches the stability issue from a model architecture perspective, our focus is on adaptive sampling techniques, highlighting a key difference. However, as suggested in this paper, it is indeed meaningful to investigate performance metrics relative to model architectures. Furthermore, it is important to study how these structures interact with adaptive sampling methods.
>
> To this end, we are currently conducting experiments to verify whether the proposed L-PINN exhibits compatibility issues with various model architectures. While the full architecture of PirateNet [1] is not yet publicly available, we are testing its main components, such as random Fourier blocks, attention mechanisms, and residual blocks, to assess their compatibility. We will report the results along with the completed experiments within a few days.
>
>
> [1] Wang, S., Li, B., Chen, Y. and Perdikaris, P., 2024. PirateNets: Physics-informed Deep Learning with Residual Adaptive Networks. arXiv preprint arXiv:2402.00326.
>
> ---
>
> > **Q1, 2**
>
> You are absolutely correct. We acknowledge the vulnerabilities in the previously derived definitions and derivations of the feature vector. To address this, we have included detailed mathematical expressions and estimation methods in **Appendix A**. Thank you for your comment; it has been invaluable in strengthening our theoretical foundation.
>
> To briefly explain the feature vector extraction process: We represent  $\mathcal{R}_{\theta}(\mathbf{x})$ as  [$g(f)$]$(\mathbf{x})$,  and apply a first-order Taylor approximation to [$g(f)$]. The Taylor expansion utilized at this point introduces the concept of the Fréchet derivative $D_g(f)$ in function spaces. The detailed construction and explanation related to this are provided in **Appendix A**.
>
> As a result, we obtained experimental outcomes that align more clearly with **Assumption 3.2** and **3.3** compared to previous derivations. Moreover, while this result represents a local approximation of feature vectors, we observed that it reasonably operates even for non-linear PDEs.

---

> > ### Comment · Reviewer_SkJ6 · 2024-11-24
> >
> > I appreciate the authors' revision. The mathematical derivation for nonlinear PDEs makes sense to me overall. However, I noted an error in Figure 6: the title of each plot contains  "layer layer."
> >
> > To further clarify the figure, I suggest adding a sentence to the caption to explain that each plot corresponds to a single PINN model with a different number of layers. Initially, I thought the figure shows a deep PINN model with visualizations of $\phi(x)$ for its hidden layers. Once this issue is addressed, I would be happy to increase my score.

---

> > > ### Author Response · Authors · 2024-11-24
> > >
> > > Dear Reviewer SKJ6,
> > >
> > > Thank you for your positive feedback! Your insightful analysis has greatly contributed to enhancing the rigor of our manuscript. Additionally, we have revised the figure and its caption in response to the confusion you pointed out and included the updated version. Once again, we sincerely appreciate your thoughtful comments and suggestions.
> > >
> > > Best regards,
> > > ICLR 2025 Conference Submission1774 Authors

---

### Official Review · Reviewer_SaWy · 2024-10-27

**Soundness:** 3
**Presentation:** 3
**Contribution:** 3
**Rating:** 8
**Confidence:** 5

**Summary:**

This paper investigates the limitations of high residual-based methods concerning learning stability as model complexity increases. The authors claim two questions which remain unclear: lack of theoretical analysis of the balancing effect and potential risks of the high residual method. They provide a theoretical analysis and propose Langevin dynamics-based PINN (L-PINN) framework for adaptive sampling of collocation points. The paper also compares the performance between L-PINN and other adaptive sampling methods.

**Strengths:**

- This paper investigates the limitations of high residual-based methods concerning learning stability as model complexity increases.

- This work provides a theoretical analysis and propose Langevin dynamics-based PINN (L-PINN) framework for adaptive sampling of collocation points.

- The paper is well-written and easy to follow.

- The figures and tables are clear.

**Weaknesses:**

- The other adaptive sampling methods (baselines) compared in the paper have some important hyperparameters which could significantly affect the performance. For example, in RAD, $k$ and $c$ are important and improper values of them will fail the method. I cannot find any description about the hyperparameters of the baseline methods. I am concerned about if the baselines are well-trained.

- It is true that applicability to large models is important for these methods. However, Applying simple 1-D PDEs to deep MLPs may not be a good choice.
  - These PDEs do not need such deep MLPs at all. Two or three hidden layers are enough for training the model.
  - Increasing the depth of MLPs will make the training unstable.
  - On the other hand, increasing width will also increase the size of the MLPs and will not make the MLPs unstable. The paper could do a comparison on this.
  - It might be more meaningful to adopt other multiple layers like attention layers and Fourier layers instead of deep MLPs. I am concerned about whether L-PINN can be applied to the scenarios that really need large models and still be robust.

**Questions:**

- “Adaptive sampling based on residual distribution” is introduced at first and then the “Adaptive sampling focused on high residuals”. Why do the authors claim the latter method is used to address convergence issues of the former method? Actually, RAR can be regarded as a special case of RAD or RAR-D. The RAD method is used to address the issue of RAR focusing too much on high residuals. RAD has less convergence issues than RAR.

- RAD methods have experimentally discussed the “Unresolved questions 2, Potential risks of the high residual method” (lines 136-139). I admit this paper discusses this problem from a different aspect. But I would like to see what is unclarified or unsolved of RAD with respect to “Unresolved questions 2”.

- A period is missing in line 148.

- In figure 4, I suggest adding standard deviations. For figure 4b, it seems L-PINN only performs better when the learning rate is 0.003. This might result from randomness. I would like to see more cases when the learning rate is between 0.002 and 0.004.

---

> ### Author Response · Authors · 2024-11-20
>
> We sincerely thank you for taking the time to review our manuscript and provide insightful comments. Your suggestions have greatly contributed to enhancing the presentation of our work. Our detailed responses are provided below.
>
> ---
>
> > **W1**
>
> We apologize for the confusion. All the settings we utilized throughout the manuscript are based on the default settings provided in the publicly available codes of the baseline papers. Additional details have been included in **Appendix F.2**.
>
> ---
>
> > **W2-1**
>
> The primary message we aimed to deliver through this paper focuses on the stability issues arising from model complexity, and most of the experiments we designed were intended to support this point. Due to this focus, we could not explore applications to more complex PDEs. However, in **Appendix I**, we discussed the compatibility of the proposed L-PINN with 2D PDEs and demonstrated its superiority over other adaptive sampling schemes.
>
> We reason that these experimental results can be explained by the following factors detailed in **Appendix I**. In summary, we attribute this to the inaccuracy of Monte Carlo integration, $\mathbb{E}|\mathcal{R}_\theta(x)|^k$, during the sampling process. This inaccuracy becomes more pronounced in higher dimensions with a limited number of collocation points. Unlike other methods that directly rely on Monte Carlo integration, L-PINN avoids this approach, contributing to its robust performance in such scenarios.
>
> > **W2-2, 3**
>
> Additional experimental results related to this issue have been included in **Appendix E.2**. Although the effect is less pronounced compared to depth, we observed similar phenomena with respect to width.
>
> > **W2-4**
>
> I also agree with that this is an important point. However, addressing more complex problems at this stage would be challenging. We believe it is crucial first to validate whether L-PINN faces compatibility issues with architectures other than MLPs. In this regard, we are conducting experiments using random Fourier blocks, modified MLP (using skip connections), and attention mechanisms. We will attach the additional results  once the experiments are completed in a few days.
>
> ---
>
> > **Q1**
>
> You are correct that the explanation might cause confusion in the flow of the text. It is true that RAR could be considered a special case of RAD and can be understood as an attempt to address its limitations. However, the criterion we aimed to distinguish was whether the algorithm fundamentally estimates the residual sampling distribution. Rather than focusing on the relationship between algorithms, we aimed to categorize them based on their operational methods. We will clarify this distinction to make the separation more apparent.
>
> ---
>
> > **Q2**
>
> We apologize for the confusion. The message we intended to convey was to emphasize the lack of theoretical reasoning for the same phenomenon. However, we agree that this overlaps with the first unresolved question. We will combine the two to present the idea more cohesively.
>
> ---
>
> > **Q3**
>
> We have identified and corrected the typographical errors.
>
> ---
>
> > **Q4**
>
> To address your point regarding the distributional characteristics of the results and the presence of randomness, we have included a boxplot for improved visualization in **Figure 4**. Furthermore, we analyzed the performance of the algorithms for learning rates between 0.002 and 0.003, as detailed in **Figure 4-(d)**. The results show that as the learning rate approaches 0.003, the algorithms begin to exhibit instability sequentially. At 0.003, all algorithms except L-PINN consistently fail to converge, and at 0.004, all algorithms fail to converge with virtually no randomness.

---

> > ### Author Response · Authors · 2024-11-25
> >
> > Dear Reviewer SaWy:
> >
> > Regarding W2-4, we have included additional experimental results in Appendix J.
> >
> > As the author-reviewer discussion period is nearing its conclusion, we kindly request you to review our responses at your earliest convenience. Should you have any additional questions or comments, we will make every effort to address them before the discussion period ends.
> >
> > We sincerely appreciate your time and valuable feedback. We look forward to hearing from you soon!
> >
> > Best regards,
> > ICLR 2025 Conference Submission1774 Authors

---

> > > ### Comment · Reviewer_SaWy · 2024-11-26
> > >
> > > Thank you for the responses. Most of my concerns has been well addressed. I am impressed by the new experiments. I have increased my score from 6 to 8.

---

> ### Author Response · Authors · 2024-11-26
>
> Dear Reviewer SaWy,
>
> We’re glad to hear that our experiments addressed your concerns. In particular, your points regarding compatibility with other model architectures and changes from the perspective of width were aspects we hadn’t considered thoroughly. Thank you for your positive feedback!
>
> Best regards,
> ICLR 2025 Conference Submission1774 Authors

---

### Official Review · Reviewer_u9zX · 2024-11-03

**Soundness:** 2
**Presentation:** 2
**Contribution:** 2
**Rating:** 5
**Confidence:** 3

**Summary:**

The paper presents a method to adaptively select the collocation points for solving partial differential equations (PDEs) through physics-informed neural networks (PINNs). In PINNs, the selection of the number and location of collocation points impacts the model training and is a well-known issue in the literature. Various methods have already been proposed in the literature, some of which have been compared in this paper. The paper presents a novel method for selecting the collocation points through empirical results and theory. The method named Langevin PINNs aims to focus not only on high-residual-based locations, but also proposes to balance it with selecting locations with low or medium residual values. Overall, the paper's motivation aligns with improving PINNs for simulating PDEs, showcasing its effectiveness on canonical examples.

**Strengths:**

The paper is easy to follow for non-theory parts and presents a concise overview of the literature in this domain.

The choice of PDEs is diverse and incorporates diverse challenges observed while training PINNs for simulating PDEs.

Sample trajectory plots help observe the method's performance.

**Weaknesses:**

The rationale for showcasing the method's performance on deep networks is unclear. The advantage gained by the algorithm with a deep neural network is unclear. From Table 1, it seems like the baselines achieve a similar result even with a small network, so why would one opt for a further deep network which is even harder to optimize?

The method is not clearly explained, and it is unclear how to implement the proposed method. It would be appreciated if the authors could provide a more detailed explanation of the method and implementation.

The discussion on the computational complexity of the baseline methods is presented. However, the manuscript does not compare the proposed method's computational cost with the baselines. Can the authors provide such a comparison of the computational cost? It would help the readers analyze the advantages of the proposed method in terms of computational cost.

Although compared with similar methods, can the authors compare their method with RAR-D presented in [1] or justify why the comparison is/should not be performed?

It is difficult to understand what Fig. 5 shows. It seems like the performance of the proposed method is completely off for a deeper network, contrary to the text in the article.

Although a sensitivity analysis of the proposed method is carried out with different learning rates, the range seems limited. How do the methods perform when trained at an even smaller learning rate?

Limitations:

The proposed method is presented for low-dimensional problems, and validating its performance on high-dimensional problems is not performed. The selection of collocation points in higher dimensions is also a complex problem. The paper does not discuss how the method scales with the rise in dimensionality.

Along similar lines, the method is not performed for multiscale systems of PDEs, which have challenges in choosing the right collocation points.

[1] Wu, Chenxi, et al. "A comprehensive study of non-adaptive and residual-based adaptive sampling for physics-informed neural networks." Computer Methods in Applied Mechanics and Engineering 403 (2023): 115671.

**Questions:**

Included along with weaknesses.

---

> ### Author Response · Authors · 2024-11-20
>
> Thank you for your thorough review and helpful suggestions. Below, we provide responses to your comments.
>
> > **W1**
>
> While shallow neural networks may achieve similar results in some scenarios, **Figure 4-(a)** shows that the relative $L^2$ error improves consistently for all sampling methods as the number of layers increases, even without learning rate decay. For L-PINN, **Appendix G** further demonstrates improved performance across most $\beta$ values with deeper architectures, suggesting that increasing layer depth is a viable strategy, even for simple 1D PDEs.
>
> Additionally, **Table 1** highlights L-PINN's stable and robust performance with deeper architectures, unlike other algorithms such as Random-R and RAD, which exhibit instability in specific cases (e.g., Allen-Cahn and KdV equations). These findings underscore the reliability of L-PINN across various PDEs without compromising stability.
>
>
> ---
>
> > **W2**
>
> To address your concern regarding implementation details, we have added Python-style pseudo code in **Appendix K** to illustrate the specific considerations made during the implementation phase. This addition aims to provide a clearer understanding of the methodology and its practical application, ensuring transparency and reproducibility.
>
> ---
>
> > **W3**
>
> To provide clarity on the computational efficiency of the proposed method, we have included a detailed analysis in **Appendix H**. In this section, we measured the time required for neural network training over 1000 iterations while varying the number of collocation points (100, 1000, 10,000, 50,000, and 100,000) and increasing the problem's dimensionality from 1D to 2D. The results are organized in a table, presenting the time taken (in seconds) for each configuration. This analysis demonstrates how the number of collocation points and problem dimensions affect the computational time during training.
>
> ---
>
> > **W4**
>
> In [1], RAR-D achieves more efficient computation cost compared to RAD by gradually concatenating sampled points to the initial random points (half of the fixed number of collocation points) rather than using the fixed number of collocation points from the beginning. However, since the initial points are not updated, this approach can result in solution error degradation compared to RAD. Furthermore, as demonstrated in [1, 2], RAD consistently outperformed RAR-D in terms of Relative $L^2$ error for solution accuracy. Based on this analysis, we believe that comparing with RAD alone sufficiently covers the performance aspects of RAR-D to a reasonable extent.
>
> [1] Wu, Chenxi, et al. "A comprehensive study of non-adaptive and residual-based adaptive sampling for physics-informed neural networks." Computer Methods in Applied Mechanics and Engineering 403 (2023): 115671.
> [2] Daw, Arka, et al. "Mitigating propagation failures in physics-informed neural networks using retain-resample-release (r3) sampling." ICML 2023.
>
> ---
>
> > **W5**
>
> Upon reviewing the issue you pointed out, we realized that the results for prediction errors were mistakenly labeled as prediction values. This has been corrected, and the updated version has been uploaded for your review.
>
> ---
>
> > **W6**
>
> The primary focus of our study is on scenarios where the learning rate is relatively high, as there is limited prior research reporting results for lower learning rates. To address this gap, we conducted experiments under lower learning rate conditions and included the findings in **Figure 4-(c)** of the manuscript. Based on the results, we observed that a learning rate of at least 0.0005 is necessary for various sampling methods to undergo a meaningful learning process.
>
> ---
>
> > **L1**
>
> Regarding the complexity of PDE problems, while it was challenging to explore a wide range of cases within the limited time frame, we conducted experiments on Burgers' 2D and Heat 2D problems and included the results in **Appendix I**. These experiments demonstrated that L-PINN outperforms other baseline algorithms in terms of performance.
>
> We reason that these experimental results can be explained by the following factors detailed in **Appendix I**. In summary, we attribute this to the inaccuracy of Monte Carlo integration, $\mathbb{E}|\mathcal{R}_\theta(x)|^k$, during the sampling process. This inaccuracy becomes more pronounced in higher dimensions with a limited number of collocation points. Unlike other methods that directly rely on Monte Carlo integration, L-PINN avoids this approach, contributing to its robust performance in such scenarios.
>
>
> ---
>
> > **L2**
>
> Our scope primarily focuses on the mathematical analysis of the weaknesses in adaptive sampling methods and the experimental validation of those findings. While we have not explored multiscale systems in this work, we acknowledge their importance and will consider further verification and improvements to extend the applicability of our approach to such systems in future research.

---

> > ### Author Response · Authors · 2024-11-25
> >
> > Dear Reviewer u9zX:
> >
> > As the author-reviewer discussion period is nearing its conclusion, we kindly request you to review our responses at your earliest convenience. Should you have any additional questions or comments, we will make every effort to address them before the discussion period ends.
> >
> > We sincerely appreciate your time and valuable feedback. We look forward to hearing from you soon!
> >
> > Best regards,
> > ICLR 2025 Conference Submission1774 Authors

---

> > > ### Comment · Reviewer_u9zX · 2024-11-26
> > > **Response to authors**
> > >
> > > Thank you for providing additional experiments and discussions. However, in my opinion, this study does not improve the current literature regarding accuracy (as shown in Table I) and computational cost (discussed in Appendix H). The provided argument that the accuracy can be increased minorly with deeper networks is not ideal for solving canonical problems. The current challenge in PINN-based approaches is accuracy and computational cost. I do not see how the proposed method contributes to alleviating these challenges. Hence, I will continue with my initial assessment.

---

> ### Author Response · Authors · 2024-11-26
>
> Dear Reviewer u9zX,
>
> We regret that we may not have fully addressed the concerns you raised. In the main text (Table 1), our primary focus was on theoretical analysis and its empirical validation, particularly emphasizing the instability that arises as model complexity increases. Regarding your concerns about accuracy and complexity, we fully understand their significance. Within the limited rebuttal period, we aimed to partially explore these aspects by including an evaluation of 2D PDE performance in Appendix I (w.r.t. accuracy), along with reasoning related to the uncertainty in MCI estimation for high-dimensional PDEs. Additionally, we highlighted that the gradient-based sampling distribution approximation, which does not rely on directly approximating MCI, is experimentally robust in terms of computational complexity with respect to $l_L$, further supporting its practical applicability.
>
> Thank you once again for pointing out these important aspects and providing constructive feedback!
>
> Best regards,
> ICLR 2025 Conference Submission1774 Authors

---

### Author Response · Authors · 2024-11-20
**Regarding the rebuttal response format**

We sincerely thank the reviewers for your detailed and insightful comments. We have made every effort to address as many of the raised issues as possible, and the updated version of the paper has been uploaded.

Our rebuttal responses reference this updated version of the paper. Consequently, the locations of specific content addressing prior concerns may have shifted slightly, and we kindly ask the reviewers to take note of this.

Additionally, we will incorporate further experimental results into the paper whenever possible before the rebuttal deadline.

---

### Comment · Area_Chair_DLKa · 2024-11-26

Dear Reviewers u9zX, SaWy,
If not already, could you please take a look at the authors' rebuttal? Thank you for this important service.
-AC

---

### Author Response · Authors · 2024-12-03
**Final Summary of the Rebuttal**

We sincerely appreciate the reviewers' insightful feedback, which has been instrumental in enhancing the quality, depth, and clarity of our work. Below, we summarize the key strengths of our approach, address the primary concerns raised, and highlight the additional efforts made during the discussion phase.

---

### Key Strengths of Our Approach

1. **Theoretical Contributions:**
   While previous studies have demonstrated the success of adaptive sampling methods in improving model performance, a rigorous theoretical framework to analyze these sampling strategies has been limited. Our work bridges this gap by proposing a detailed theoretical analysis that examines sampling methods in relation to model complexity, residual concentration, and stability. This framework not only provides insights into existing methods but also lays the groundwork for advancing adaptive sampling strategies in PINNs.

2. **Novelty:**
   By integrating Langevin dynamics into Physics-Informed Neural Networks (PINNs), our approach introduces a novel mechanism that fundamentally enhances learning stability. Unlike existing methods that rely on Monte Carlo integration, our method avoids direct reliance on these techniques, which makes it inherently more robust when applied to higher-dimensional PDEs. This innovation addresses critical challenges in scaling PINNs to complex problems.

3. **Comprehensive Validation:**
   To validate the proposed theoretical framework, we designed experiments explicitly aligned with the analysis presented in the manuscript. These experiments evaluate key aspects, including the impact of model complexity, residual concentration, and adaptive sampling on learning stability. The results not only reinforce the theoretical insights but also demonstrate the robustness and generalizability of L-PINN across a wide range of PDE scenarios.

---

### Addressing Reviewer Concerns

1. **Mathematical Clarification of Feature Vectors:**
   To address feedback regarding feature vector construction, we revisited the mathematical formulation and clarified the derivation process in **Appendix A**. Specifically, we refined the representation of the feature vectors to better align with assumptions. This update resolves ambiguities and ensures consistency with the proposed theoretical framework, strengthening the connection between theory and practice.

2. **Scalability to Complex Scenarios:**
   To address concerns regarding scalability, we conducted new experiments on two-dimensional PDEs, including Burgers' and Heat equations, which are outlined in **Appendix I**. Additionally, we performed a computational complexity analysis, detailed in **Appendix H**, to evaluate the efficiency of L-PINN in complex scenarios (more collocation points, higher dimension). These results highlight L-PINN's ability to efficiently scale to complex scenarios without compromising stability or performance.

3. **Comparative Analysis with Alternative Architectures:**
   In response to reviewer feedback, we expanded our comparative analysis to include alternative architectures, such as random Fourier features, attention mechanisms, and modified MLPs (e.g., incorporating residual blocks) detailed in **Appendix J**. These additional experiments confirmed that L-PINN maintains compatibility and effectiveness across diverse model structures, further underscoring its adaptability and versatility.

4. **Clarification of Explanations:**
   We recognize the ambiguities in certain parts of our manuscript, particularly regarding distinctions between methods like RAD and RAR-D and their theoretical implications. Additionally, we acknowledge that the interpretation of **Figure 4-(a)** may have introduced confusion. While immediate revisions are not feasible, we will address these points in the final version by clearly separating and structuring the relevant discussions and improving the explanation of experimental results, especially those in Figure 4-(a). These refinements aim to enhance the clarity and accessibility of our explanations.

---

### Final Remarks

The reviewers' constructive feedback has been invaluable in improving our manuscript and resolving potential ambiguities. Through additional experiments, expanded comparative analyses, and planned improvements to explanations, we have strengthened both the theoretical and experimental foundation of L-PINN.

We believe that our work offers a meaningful contribution to the field by providing a robust framework that improves learning stability in PINNs through balanced sampling strategies. Once again, we sincerely thank the reviewers for their thoughtful comments and constructive suggestions, which have greatly enriched our work.

---

### Meta-Review · Area_Chair_DLKa · 2024-12-20

**Metareview:**

This paper considers improving PINN, which uses a neural network to represent the solution to a PDE and seek it by minimizing an integration of the PDE residual. Accurately evaluating the integrated residual is an outstanding challenge, because PINN is believed to particularly suit high dimensional problems, for which full integrations are however too expensive to perform. The authors proposed a Langevin SDE based approach to randomly sample collocation points for evaluating the residual in a balanced fashion, so that high residual locations are emphasized but low residual locations are not ignored either. Reviewers and I all agree this is an interesting idea. However, there were concerns about robustness / details of the implementation and inadequate evidence of improved performance over existing approaches, which remained unresolved after the rebuttal. Moreover, although the sampling problem matters more in high dimensions, only low dimensional PDEs were tested in the paper. In addition, there was insufficient comparison with non-PINN approaches. Therefore, I encourage the authors take these discussions into consideration and submit again.

**Additional Comments On Reviewer Discussion:**

Reviewer u9zX was concerned with the inadequate evidence of improved performance over existing approaches, which was not fully resolved post rebuttal.
Reviewer Zj8i was concerned with robustness / details of the implementation, which was not fully resolved post rebuttal.
Moreover, although the sampling problem matters more in high dimensions, only low dimensional PDEs were tested in the paper. In addition, there was insufficient comparison with non-PINN approaches.

---

### Decision · Program_Chairs · 2025-01-22

Reject